# Estimating snow accumulation and ablation with L-band InSAR

Jack Tarricone[1,2], Ryan W. Webb[3], Hans-Peter Marshall[4], Anne W. Nolin[2,1], and Franz J. Meyer[5]

[1]Graduate Program of Hydrologic Sciences, University of Nevada, Reno, Reno, NV, USA
[2]Department of Geography, University of Nevada, Reno, Reno, NV, USA
[3]Department of Civil & Environmental Engineering & Construction Management, University of Wyoming, Laramie, WY, USA
[4]Department of Geosciences, Boise State University, Boise, ID, USA
[5]Geophysical Institute, University of Alaska Fairbanks, Fairbanks, AK, USA

**Correspondence:** Jack Tarricone (jtarricone@nevada.unr.edu)

**Abstract.** Snow is a critical water resource for the western US and many regions across the globe. However, our ability to accurately measure and monitor changes in snow mass from satellite remote sensing, specifically its water equivalent, remains a challenge. To confront these challenges, NASA initiated the SnowEx program, a multi-year effort to address knowledge gaps in snow remote sensing. During SnowEx 2020, the Uninhabited Aerial Vehicle SAR (UAVSAR) team acquired an L-band Interferometric Synthetic Aperture Radar (InSAR) data time series to evaluate the capabilities and limitations of repeat-pass L-band InSAR for tracking changes in snow water equivalent (SWE). The goal was to develop a more comprehensive understanding of where and when L-band InSAR can provide SWE change estimates, allowing the snow community to leverage the upcoming NASA-ISRO SAR (NISAR) mission. Our study analyzed three InSAR image pairs from the Jemez Mountains Basin, NM, between 12–26 February 2020. We developed a snow-focused multisensor method that uses UAVSAR InSAR data synergistically with optical fractional snow covered area (fSCA) information. Combining these two remote sensing datasets allows for atmospheric correction and delineation of snow covered pixels within the radar swath. For all InSAR pairs, we converted phase change values to SWE change estimates between the three acquisition dates. We then evaluated InSAR-derived retrievals using a combination of fSCA, snow pits, meteorological station data, in situ snow depth sensors, and ground-penetrating radar (GPR). The results of this study show that repeat-pass L-band InSAR is effective for estimating both snow accumulation and ablation with the proper measurement timing, reference phase, and snowpack conditions.

## 1   Introduction

### 1.1   Significance and Motivation

In the western US (WUS), seasonal mountain snowmelt produces approximately 70 % of the annual discharge (Li et al., 2017a), and is the primary water source for about 60 million people (Stewart et al., 2004). To adequately manage this resource, an accurate accounting of the spatiotemporal variations in snow water equivalent (SWE) is needed (Bales et al., 2006). Climate change is affecting the stationarity of the WUS hydrologic cycle (Milly et al., 2008), causing an overall decline in mountain

snowpack (Mote et al., 2018), and emphasizing the importance of properly monitoring snow into the future (Siirila-Woodburn et al., 2021).

Water managers could benefit from regular repeat coverage of spatially distributed, low-latency SWE data at spatial resolutions that are appropriate for mountain water forecasting. While remote sensing has made significant advances in measuring snow properties, there is still no remote sensing technique that can continually measure SWE from space for mountain hydrologic applications (Lettenmaier et al., 2015). Here, we explore L-band InSAR for monitoring changes in SWE.

## 1.2  Background and Previous Work

The most effective and widely used SWE estimation technique combines suborbital lidar (Deems et al., 2006; Trujillo et al., 2007) with hyperspectral imaging (Nolin et al., 1993) to produce both snow depth and fractional snow covered area (fSCA) at the watershed scale. Converting these measurements into SWE requires spatially distributed snowpack energy balance modeling (Painter et al., 2016). Like all optical techniques, lidar and hyperspectral imaging are limited by cloud cover, which can be frequent in mountain environments, and global spaceborne monitoring with lidar is not currently practical.

Since the 1970s, spaceborne passive microwave radiometers have used brightness temperature to estimate SWE at hemispherical scales (Rango et al., 1979). More recent studies utilized the Advanced Microwave Scanning Radiometer-EOS (ASMR-E) and the Scanning Multichannel Microwave Radiometer (SMMR) to continue the development of this SWE estimation technique (Derksen et al., 2002; Vuyovich et al., 2014). These instruments produce data on the spatial scale of tens of kilometers, limiting their ability to capture the topographic and snowpack heterogeneity of mountain environments. Passive microwave retrievals are also limited to dry snowpacks < 1 m of snow depth due to signal saturation (Foster et al., 2005).

While passive microwave remote sensing is not well suited for mountain environments, active microwave (radar) has shown promise for snowpack monitoring. Time-of-flight approaches have been used for decades from ground-based (Gubler and Hiller, 1984; Marshall and Koh, 2008) and airborne (McGrath et al., 2018; Lewis et al., 2017) platforms. Synthetic aperture radar (SAR) is an active microwave remote sensing technique that addresses the two main deficiencies in both optical and passive microwave; it can penetrate through clouds and has a spatial resolution on the scale of tens of meters instead of kilometers.

Spaceborne applications of SAR for estimating snow properties have mostly focused on backscatter approaches, where shorter wavelengths (Ku- and X-band) have been used to estimate SWE (Rott et al., 2010; Yueh et al., 2009; King et al., 2018; Zhu et al., 2021). However, this method requires a complex dense media radiative transfer model (DMRT) with input parameters that not only include snow density ($\rho_s$) and snowpack liquid water content (LWC), but also parameters such as snow stratigraphy, snow grain size, and ground surface conditions. Snow microstructure parameters are challenging to precisely estimate over large spatial scales (Rutter et al., 2019).

SAR is proven for measuring snow wetness (Nagler and Rott, 2000; Nagler et al., 2016; Lund et al., 2020) as wet snow attenuates the radar signal, causing a decrease in backscatter intensity when compared to dry snow conditions. New backscatter methods are being developed to measure snow depth at C-band (Lievens et al., 2019, 2022). This technique shows promise, especially in deeper snowpacks (> 1 m), but the underlying physics governing the retrievals are not yet well characterized.

Recently, the use of Interferometric Synthetic Aperture Radar (InSAR) to estimate SWE has become an area of interest because of the higher temporal (12 days) frequency and L-band (~24 cm) wavelength of the future NASA-ISRO SAR (NISAR) mission (Rosen et al., 2017). InSAR uses the differences in radar phase between subsequent overpasses to estimate surface displacement. The InSAR SWE theory, initially proposed by Guneriussen et al. (2001), relates changes in the interferometric phase of a radar signal to SWE changes of dry snow on the ground between acquisitions.

A series of studies have shown the further utility of these InSAR methods for snow, such as Rott et al. (2003) in Austria, and Deeb et al. (2011) on Alaska's north slope using European Remote-Sensing Satellite (ERS-1) C-band radar. Leinss et al. (2015) conducted an intensive season-long ground-based dual-frequency (Ku- and X-band) interferometric experiment in Finland with measurements every four hours, where they found the method was successful for continually measuring SWE in dry taiga snow, but that liquid water and vegetation quickly cause coherence loss at these higher frequencies.

More recent studies have also used C-band radar from various space-borne platforms. The Sentinel-1 A/B radar was utilized in Finland, leveraging the more consistent overpass repeat cycle (Conde et al., 2019). Li et al. (2017b) analyzed two InSAR pairs from the Envisat ASAR instrument in the Tianshan Mountains of northwestern China, where they found promising results but were limited by large interferometric temporal baselines and the lack of in situ validation data. Eppler et al. (2022) used a nine-year RADARSAT-2 time series in Canada to develop "SlopeVar", a method for estimating SWE change without phase unwrapping by spatially correlating phase sensitivity to local topography. Nagler et al. (2022) conducted an airborne L- and C-band experiment in the Austrian Alps in preparation for Radar Observation System for Europe in L-band (ROSE-L). While their results are preliminary, they show good performance for tracking snowfall events at L-band because of its lack of impairment from $2\pi$ phase wrapping ambiguities.

These orbital InSAR studies showed promise for estimating SWE but lacked sufficient temporal length and variety of vegetation, topography, and snowpack characteristics. Moreover, they also lacked adequate validation data and a small spatial scale to thoroughly understand the technique's limitations and synergies with other types of snow measurements.

## 1.3   Research Objectives

To address these InSAR-derived SWE limitations, the 2020 NASA SnowEx campaign (Marshall et al., 2019) conducted an Uninhabited Aerial Vehicle Synthetic Aperture Radar (UAVSAR) L-band InSAR time series flight campaign at 13 research sites across the WUS. The goal of the 2020 SnowEx experiment was to test L-band InSAR's ability to measure SWE changes in a wide range of geographic locations, snow conditions, and land cover types with corresponding in situ ground-based observations. InSAR-derived snow depth changes measured over a two-week interval on the open western end of Grand Mesa, CO in February 2020 showed high correlation ($r^2$ = 0.76) with snow depth differences measured by coincident repeat lidar from the same time period. RMSE differences between the two 5 m resolution depth change maps were within typical lidar error ($< 5$ cm) for depth and 0.9 cm of SWE (Marshall et al., 2021).

The overall goal of this study is to assess the performance of L-band InSAR for monitoring SWE changes in an environment where there is both snow accumulation and ablation (melt, evaporation, or sublimation). Currently, this UAVSAR-based approach has only been applied to cold dry snow conditions on Grand Mesa (Marshall et al., 2021), where the snow depth

variations were mainly driven by wind redistribution, but not melt or evaporation. Towards this end, the specific objectives of the work presented here are to (1) analyze InSAR SWE retrievals over a complex mountain region and (2) validate the retrievals using satellite and in situ data.

## 2 Methods

To achieve our objectives, we analyzed three interferometric image pairs that were acquired over the Jemez Mountains, New Mexico. First, we developed a workflow (Tarricone, 2023) that (a) corrects the observed interferometric phase for atmospheric delay and (b) corrects incidence angle error effects by using improved incidence angle estimates derived from airborne lidar. We then computed spatial changes in SWE over the study area and evaluated our SWE retrievals using fSCA, ultrasonic snow depth sensors, ground-penetrating radar (GPR), and snow pits.

Section 2 (Methods) is split into the following subsections: 2.1 overviews InSAR for estimating SWE changes, 2.2 describes the study area, 2.3 reviews the remote sensing and in situ data, 2.4 is a description of the atmospheric correct steps, 2.5 explains the creation of new incidence angle data, and 2.7 outlines the SWE change calculation.

### 2.1 InSAR for detecting SWE changes

InSAR is an active remote sensing technique that uses the differences in phase to map surface topography (single-pass) (Zebker and Goldstein, 1986) or various types of surface deformation (repeat-pass) (Goldstein and Zebker, 1987). Using the precise location of the orbit or flight pattern, the phase difference between the two (repeat-pass) acquisitions can be used to calculate deformation on the centimeter scale. Traditionally repeat-pass InSAR (Rosen et al., 2000), where the sensor scans the same area at two different times, has been used to monitor tectonic motion (Funning et al., 2005), geomorphic processes (Colesanti et al., 2003), ice sheet velocity (Mouginot, 2012), and volcanic activity (Poland and Zebker, 2022).

For snow applications, Guneriussen et al. (2001) theorized a relationship between InSAR phase change and change in dry SWE between acquisitions. Dry snow has low attenuation of the radar signal, and at frequencies below 10 GHz (Marshall et al., 2005; Ulaby et al., 1984), the majority of the backscatter stems from the snow ground interface. Dry snow and the atmosphere have different dielectric properties, causing a refraction or directional change of the radar propagation path and a decrease in speed when the signal propagates through the snow layer (Figure 1). The refraction and wave speed are controlled by the refractive index of snow, which is governed by $\rho_s$. We leverage these previous studies to develop a current workflow applied to UAVSAR data acquisitions.

To isolate the SWE change impacts on the phase, other factors impacting phase must be identified and compensated for. Outlined in Deeb et al. (2011) and updated for suborbital acquisition considerations, total interferometric phase includes the following contributions:

$$\phi_{total} = \phi_{flat} + \phi_{topo} + \phi_{atm} + \phi_{snow} + \phi_{random} + \phi_{systematic} \tag{1}$$

where $\phi_{flat}$ and $\phi_{topo}$ are phase impacts from the flat Earth and local topography, which are both accounted for in the UAVSAR InSAR processing chain using the Shuttle Radar Topography Mission (SRTM) DEM as input. $\phi_{random}$ is the random error, where the majority comes from temporal decorrelation (Zebker et al., 1997). $\phi_{systematic}$ represents the systematic error within the UAVSAR instrument. This error is mainly associated with uncertainty in the plane's position and deviations in the flight track between acquisitions. Variations in the plane's position are accounted for within the UAVSAR processing workflow as best as possible, but not all aircraft motion can be completely captured, which can leave residual phase change.

Assuming all previously mentioned errors are accounted for, extracting $\phi_{snow}$ from the observed phase ($\phi_{total}$) in UAVSAR data mostly requires an accurate compensation of $\phi_{atm}$, which is the phase contribution from change in path delay through the atmosphere. Refer to Subsection 2.4 for a detailed explanation of how $\phi_{atm}$ is addressed in our approach. Once $\phi_{snow}$ is isolated, the measured phase shifts are used to estimate SWE using the following equation proposed by Guneriussen et al. (2001), which accounts for both the path length change caused by refraction and the change in wave speed in snow:

$$\Delta\text{SWE} = -\frac{\Delta\phi_{snow}\lambda}{4\pi} \cdot \rho_s \cdot \frac{1}{\cos\theta - \sqrt{\epsilon_s - \sin^2\theta}} \tag{2}$$

where $\Delta\text{SWE}$ is the change in SWE between acquisitions, $\lambda$ is the radar wavelength (23.84 cm for UAVSAR), $\theta$ is the radar incidence angle, and $\epsilon_s$ is the real part of the dielectric permittivity of snow. For dry snow, there is a direct relationship between $\epsilon_s$ and $\rho_s$, whereas for wet snow, the relationship becomes more complex, with even small amounts of liquid water vastly increasing $\epsilon_s$ values. Recent studies from Eppler et al. (2022) and Leinss et al. (2015) found that error in density estimates only biases total SWE change by $<$~5 % for dry snow in a wide range of $\theta$ ($< 50°$) and $\rho_s$ ($< 500$ kg m $^{-3}$). Leinss et al. (2015) also showed a nearly linear relationship between $\Delta\text{SWE}$ and interferometric phase for dry snow, which simplifies the SWE estimation. That said, we used Equation 2 because our study considers melting snow and $\epsilon_s$ is a direct input.

## 2.2 Description of the Study Area

Located in northern New Mexico, U.S.A., the Jemez Mountains and Jemez River Basin are on the southern extent of the Rocky Mountains (Figure 2b). The UAVSAR swath encompasses portions of Valles Caldera National Preserve (VCNP) (35°53'N, 106°32'W) (Figure 2a). This area is mainly a mountain conifer forest environment consisting of Douglas fir, white fur, and blue spruce. VCNP is surrounded by lower elevation semi-arid desert. Within the swath also lies the Valles Caldera, a 25 km wide volcanic structure dating back about 1.2 million years. Within VCNP is Valle Grande(VG) (Figure 2f), an extensive open grassland. Many resurgent lava domes form peaks over the grassy valleys, the highest of which is Redondo Peak (3430 m). About 50 % of the total annual precipitation falls in the summer months as rain from convective monsoonal storms, and the rest falls in the winter as snow. The water in this area drains into the East Fork of the Jemez River and eventually to the Rio Grande. The nearby Quemazon Natural Resource Conservation Services (NRCS) Snow Telemetry (SNOTEL) site (35°55'N, 106°24'W, 2898 m) has a 1980–2022 average peak SWE of 22.4 cm.

We focus our analysis on an 82.5 km$^2$ section of the UAVSAR swath that encompasses VG and the surrounding forested hillslopes. The study area is defined by the red rectangle in Figure 2a, with inset maps showing elevation (Figure 2d), slope

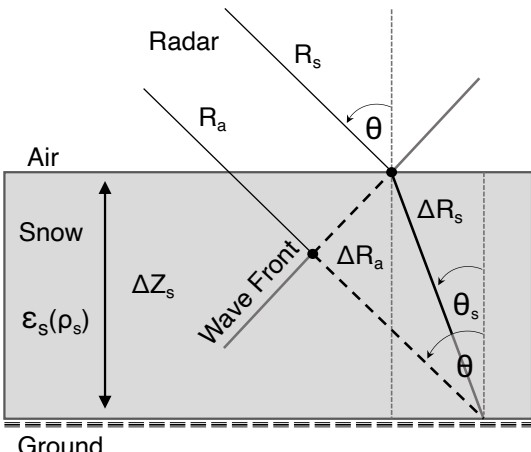

**Figure 1.** Diagram adapted from Leinss et al. (2015) showing the geometric principle of the InSAR SWE retrieval. $R_a$ represents propagation through atmosphere (no snow) and $R_s$ (with snow) to the wave front. The amount of refraction ($\theta_s$) and change in wave speed are controlled by $\epsilon_s$, which is a function of snow $\rho_s$. The variation in path length with and without snow is equal to $\Delta R_s$ - $\Delta R_a$. This path length difference causes a phase delay which is used to estimate SWE changes.

(Figure 2e), binned north and south aspects with VG delineated (Figure 2f), and 2016 NLCD canopy cover percentage (Figure 2g).

## 2.3 Data Description

### 2.3.1 UAVSAR

UAVSAR is a fully polarimetric L-band radar deployed on a NASA Gulf Stream III aircraft, traditionally flown at ~13,700 m with a 22 km nominal swath width (Hensley et al., 2008; Rosen et al., 2006). Detailed technical specifications of the radar are provided at the top of Table 1. UAVSAR data were accessed using the Python package uavsar_pytools (Keskinen and Tarricone, 2022). It uses the asf_search API (https://github.com/asfadmin/Discovery-asf_search) for easier downloading, 160 formatting, and analysis of UAVSAR data. The flights used in this study occurred on the mornings of 12, 19, and 26 February 2020. The UAVSAR team at the NASA Jet Propulsion Laboratory (JPL) processed two 7-day (12–19 and 19–26 February) and one 14-day (12–26 February) ground projected (GRD) InSAR pairs. They were unwrapped using the Integrated Correlation and Unwrapping (ICU) algorithm (Goldstein et al., 1988). Processing parameters are outlined at the bottom of Table 1, and information about the specific products used is provided in the Supplement. For the three flights used in this study, the flight 165 track baseline was maintained within $< \pm$ 3 m, which is within the $< \pm$ 5 m requirement (Hensley et al., 2008).

Three interferometric products: coherence, unwrapped phase, and the interferogram are produced for each InSAR pair. Coherence measures the consistency of the scattering characteristics within a pixel between InSAR acquisitions. Unwrapped

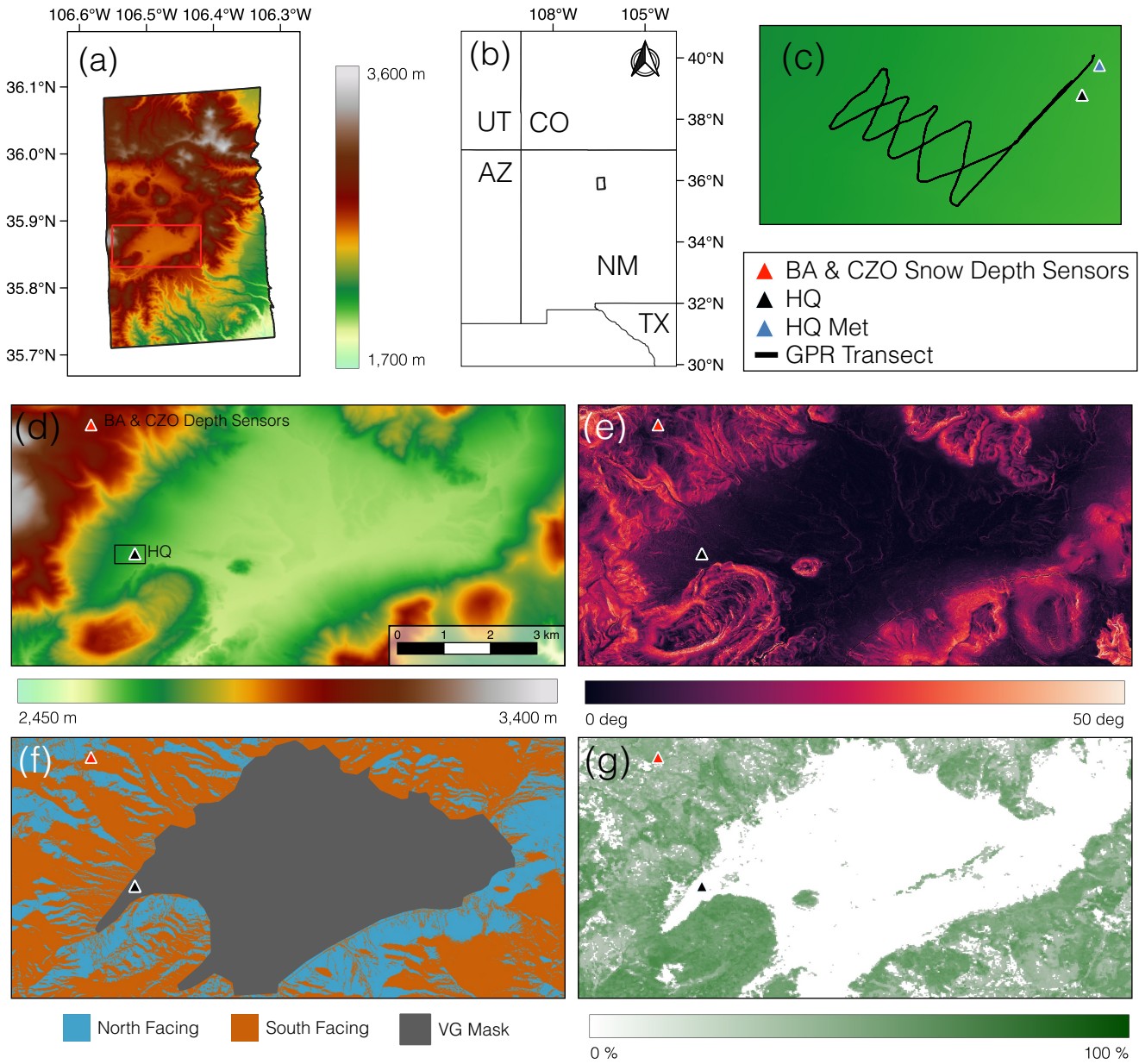

**Figure 2.** (a) DEM of the UAVSAR acquisition area provided by NASA, with a red rectangle outlining the study area. (b) Map showing the area of the UAVSAR acquisition (black outline) in the Jemez Mountains, NM. (c) A close-up of the GPR transect outlined by the black rectangle in (d), with the HQ Met (blue triangle) and HQ snow pit (black triangle) displayed. Due to their close proximity, a single red triangle represents the BA pit and CZO snow depth sensors. Within the study area extent: (d) lidar DEM, (e) lidar-derived slope, (f) lidar aspect binned to north (270-90°) (blue) and south (90-270°) (orange) facing slopes, with the gray area representing the flat VG meadow where aspect values are not valid, and (g) NLCD canopy cover percentage.

**Table 1.** Technical Specifications of the UAVSAR L-band radar (top). InSAR processing and data parameters (bottom).

| Parameter | Value |
|---|---|
| Wavelength | 23.84 cm |
| Frequency | 1.26 GHz |
| Polarization | Quad Pol |
| Bandwidth | 80 MHz |
| Pulse Length | 40 $\mu$s |
| Radar Look Direction | Left |
| Range Swath Width | 22 km |
| Average Near Range Look Angle | 28.01° |
| Average Far Range Look Angle | 68.9° |
| Ground Range Pixel Spacing | 6 m |
| Number of Looks in Range | 3 |
| Number of Looks in Azimuth | 12 |
| Phase Unwrapping Method | ICU |
| Phase Unwrapping Filtering Method | Low Pass |
| Phase Unwrapping Filter Window Size | 3 × 3 pixels |

**Table 2.** UAVSAR unwrapped phase (UNW) and coherence statistics for the full scene (FS) and study area (SA). UNW Loss (%) is the percentage of pixels lost in the unwrapping process.

| Pair | Polarization | FS Mean Coherence | SA Mean Coherence | FS UNW Loss (%) | SA UNW Loss (%) |
|---|---|---|---|---|---|
| 12–19 Feb. | HH | 0.53 | 0.50 | 9.4 | 7.7 |
| 12–19 Feb. | VV | 0.54 | 0.50 | 8.9 | 12.8 |
| 19–26 Feb. | HH | 0.55 | 0.52 | 5.0 | 4.1 |
| 19–26 Feb. | VV | 0.57 | 0.54 | 4.3 | 3.7 |
| 12–26 Feb. | HH | 0.50 | 0.48 | 8.3 | 6.6 |
| 12–26 Feb. | VV | 0.53 | 0.51 | 6.7 | 5.5 |

phase is the estimated absolute phase change in a pixel, generated from the initially ambiguous interferogram, which is defined modulo $2\pi$.

When there is a significant change in the landscape scattering properties between InSAR acquisitions, phase noise and fringe discontinuities increase, coherence decreases, and the unwrapping algorithm performs less reliably (Balzter, 2001). We analyzed the coherence and unwrapped phase products for the HH and VV polarizations to assess their quality before

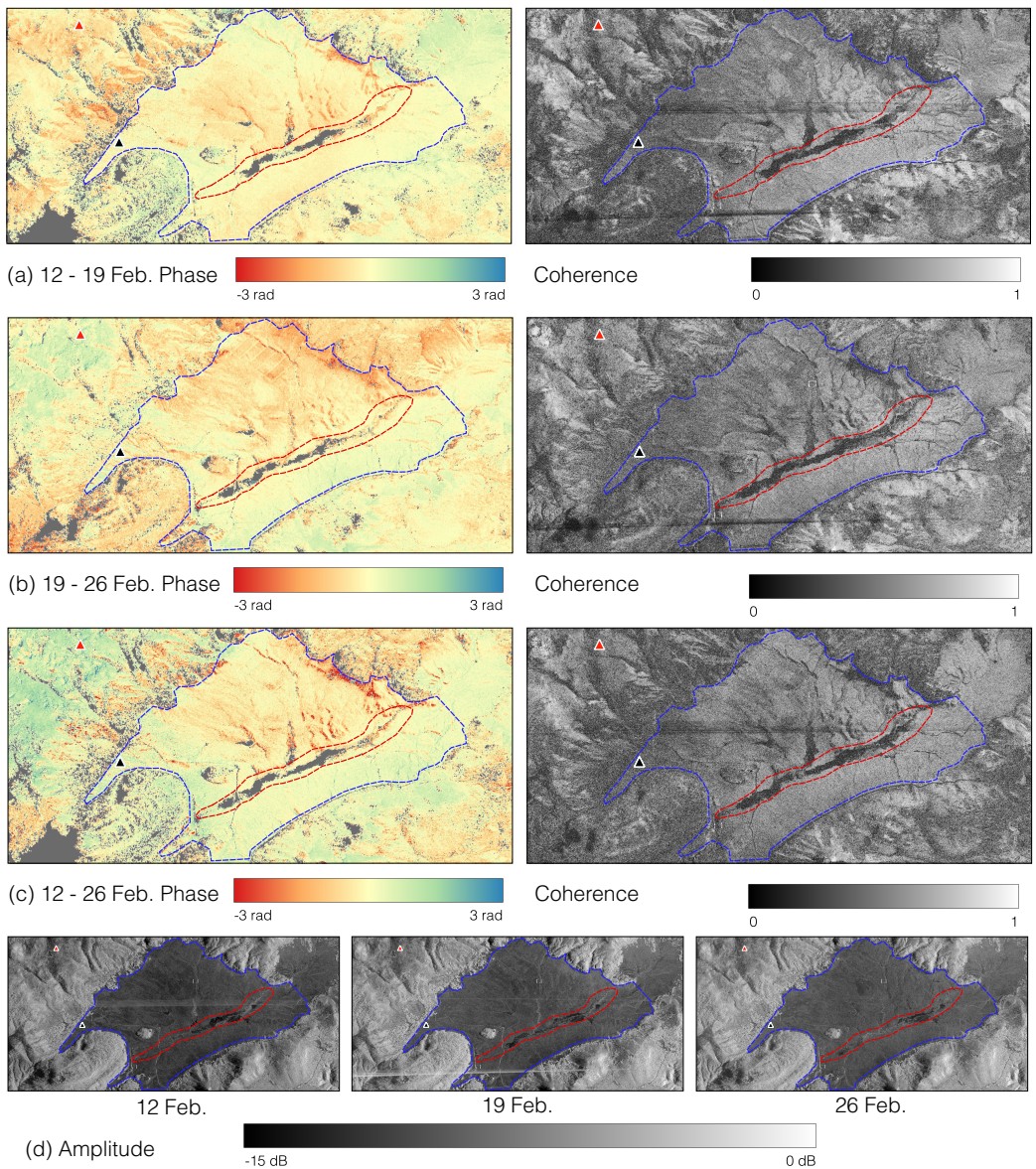

**Figure 3.** The unwrapped phase and coherence data for the (a) 12–19 February, (b) 19–26 February, and (c) 12–26 February InSAR pairs. The amplitude data (d) for the three UAVSAR flights. Both (a) and (c) were atmospherically corrected. The gray area in the phase data are pixels lost in the unwrapping processes. VG and Jemez River main channel are outlined by blue and red dotted lines, respectively. Triangles show the BA (red) and HQ (black) pits.

implementing the SWE change equation. We found co-polarizations performed similarly (Table 2), and choose to utilize HH for our study.

The right side of Figure 3 shows the coherence values in the study area for (a) 12–19 February, (b) 19–26 February, and (c) 12–26 February. In the coherence maps, there are clear patterns with respect to topography and probable variations in snowpack LWC. The area of lowest coherence surrounds the main channel of the Jemez River (red dotted line) that flows through the south-central portion of VG. As seen in Figure 3d there is a variable area of low backscatter in all three amplitude images. This backscatter decrease is likely caused by snowpack LWC or subnivean surface water attenuating the radar signal. The spatial

variability in backscatter values in this riparian area between acquisitions causes low coherence and the loss of pixels in the unwrapping processes for all three pairs. There are also horizontal streaks of low coherence and high backscatter within the images. These are likely a result of radio frequency interference (RFI) during the acquisitions. These lines do not propagate into the unwrapped phase data and therefore are not of concern.

     The left side of Figure 3a–c displays the unwrapped phase values for the three InSAR pairs. The 12–19 February and 12–

26 February pairs show the atmospherically corrected data, with this methodology discussed in Section 2.4. In the open VG meadow (blue dotted line), the unwrapping algorithm performs well, and most pixels are preserved except in the riparian area for the reasons described previously. The other source of low coherence and corresponding unwrapping pixel loss occurs on the forested hill slopes (outside of the blue dotted line) surrounding the VG meadow. Overall, the unwrapped data provides a high-quality input into the SWE change inversion equation.

### 2.3.2   Landsat fSCA

No current technique can confidently discriminate dry snow cover using solely L-band radar (Tsai et al., 2019). Our study aims to assess the ability of L-band InSAR to estimate spatiotemporal SWE changes. Therefore, our analysis requires properly identifying snow covered pixels within the UAVSAR swath, ensuring the radar signal interacts with mostly snow cover and not bare ground. To do this, we utilized Landsat 8 fSCA (U.S. Geological Survey and Center, 2018) data from 18 February

and 5 March 2020 (Figure 4). These data are generated using a spectral unmixing analysis based on the Snow Covered Area and Grain size (SCAG) algorithm developed for MODIS (Painter et al., 2009). The data processing workflow includes water masking, cloud masking, and canopy cover corrections (Selkowitz et al., 2017; Stillinger et al., 2023). Within the full UAVSAR swath, 29.7 % of pixels were entirely snow free on 18 February (Figure 4a), increasing to 38.1 % on 5 March (Figure 4b). For just the study area, 4.1 % of pixels were snow free on 18 February (Figure 4d), with an increase to 9.1 % by 5 March (Figure

4e).

### 2.3.3   Snow Pit and Meteorologic Station Data

Snowpack information was collected at two pit locations during each of the three UAVSAR overflights. These data are stored in the SnowEx database (Johnson and Sandusky, 2023). The Headquarters Meteorologic station (HQ Met) pit was located at 35°51'30"N, 106°31'17"W, at an elevation of 2650 m. The Burned Area (BA) pit was located near Redondo Peak at

35°53'18"N, 106°31'57"W, at an elevation of 3030 m.

     Measurements of snow depth, snow layer stratigraphy (grain size, grain shape, hand hardness, and manual wetness), $\rho_s$, $\epsilon_s$, and temperature were recorded for each pit. $\rho_s$, $\epsilon_s$, and temperature were measured in 10 cm increments starting at the top

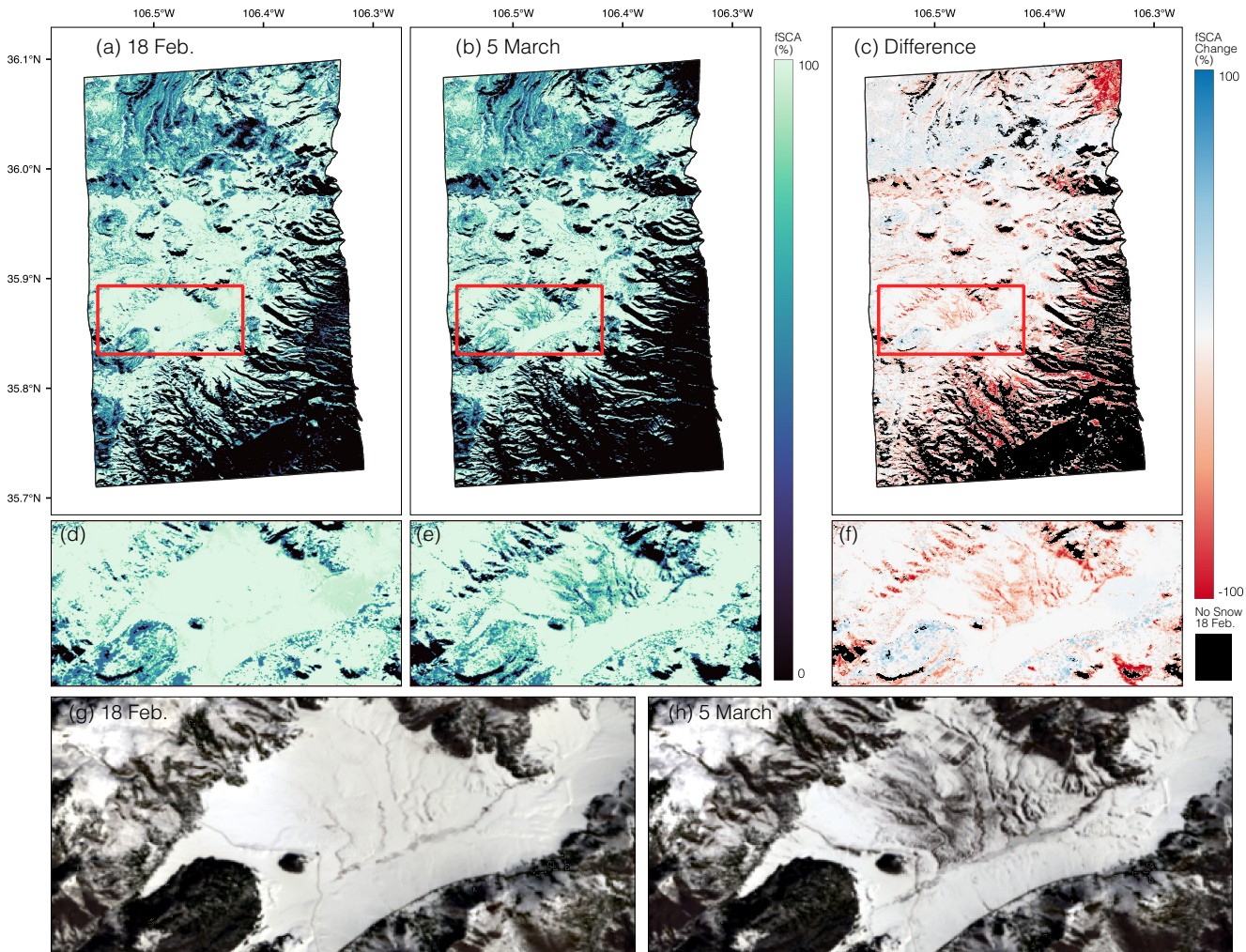

**Figure 4.** Landsat fSCA clipped to the UAVSAR swath extent (black outline) for (a) 18 February 2020 and (b) 5 March 2020. (c) The pixel-wise percent fSCA change between the two dates, with the black area representing 0 % fSCA from 18 February 2020. The study area fSCA (red box) for (d) 18 February 2020, (e) 5 March 2020, (f) and the difference between the two dates. Landsat true color image in the study area for (g) 18 February 2020 and (h) 5 March 2020.

of the pit. Stratigraphic layer size is variable and defined by the observer. In situ $\rho_s$ measurements have been shown to have an uncertainty of ~10 % (Conger and McClung, 2009; Proksch et al., 2016). $\epsilon_s$ was measured using an A2 Photonics WISe instrument (A2P, 2021), which Webb et al. (2021b) showed to have a mean absolute error (MAE) of 0.106 when compared to other in situ observations.

Summary statistics from each pit are located in Table 3. Interval boards, which are small plastic manual precipitation gauges placed on the snow surface used to track new snow accumulation, were located in close proximity to both snow pits. The HQ

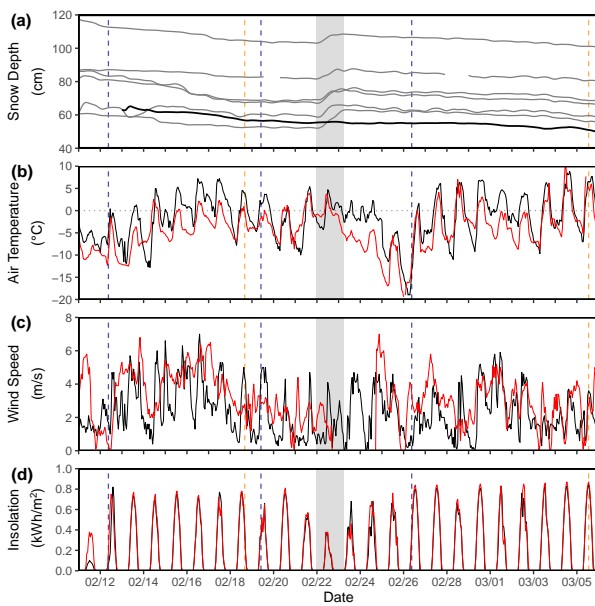

**Figure 5.** (a) A snow depth time series of the six CZO snow depth sensors (gray lines) (~3030 m) and HQ Met (black) (2650 m). The gray-shaded area represents a small storm registered by the sensors on Redondo Peak. HQ Met and RP Met (red) (3231 m) time series of (b) average hourly temperature, (c) average hourly wind speed, (d) and average hourly incoming solar radiation (insolation) from 11 February to 6 March. The vertical blue dotted lines represent the three UAVSAR flights (12, 19, 26 February), and the vertical orange dotted lines represent the Landsat fSCA acquisitions (19 February and 5 March).

pit data noted minor melting for both the 20 and 26 February, and it is important to clarify that the snow pits were collected
~1–3 hours after the radar data acquisition.

The Western Climate Research Center (WRCC) deployed two meteorologic stations that measured snow depth (Figure 5a), air temperature (Figure 5b), wind speed (Figure 5c), and incoming solar radiation (Figure 5d). The first station is the aforementioned HQ Met, and the second is located on Redondo Peak (RP Met) (35°53'02"N, 106°33'13"). Six ultrasonic snow depth sensors (~3030 m) originally installed by Molotch et al. (2009) and used in subsequent studies (Musselman et al.,
2008; Harpold et al., 2015), were used to measure variations in snow depth near the BA pit. Ultrasonic snow depth sensors have a known uncertainty of $\pm$ 1 cm (Ryan et al., 2008).

Figure 5 is a time series of in situ meteorologic data from HQ Met and RP Met. Figure 5a shows snow depths from seven ultrasonic sensors from 11 February to 6 March. The gray-shaded area on the plot shows a small snowfall event starting the night of 22 February and ending 23 February. In situ snow depths were converted to SWE by multiplying by the bulk $\rho_s$ (from
snow pit observations) for the 12–19 and 12–26 February pairs. We used a $\rho_s$ of 240 kg m$^{-3}$ for new snow measured from the BA pit interval board for the 19–26 February pair.

**Table 3.** Snow pit data collected for the UAVSAR time series. Bulk $\rho_s$ and $\epsilon_s$, which is an average of the 10 cm segments, are reported. No $\epsilon_s$ was collected at the BA pit. Data were collected on 20 February and not during the 19 February flight date. No BA pit was dug on February 12. The 12 February SWE was estimated using of 10 depth measurements around the pit area and the 20 February $\rho_s$.

| Pit | Date | UAVSAR start (HHMM LT) | Pit start (HHMM LT) | Depth (cm) | Bulk $\rho_s$ (kg m$^{-3}$) | SWE (cm) | Bulk $\epsilon_s$ | Condition |
|-----|------|------------------------|---------------------|------------|------------------------------|----------|-------------------|-----------|
| HQ | 2/12 | 0946 | 1305 | 78 | 261 | 20.3 | 1.26 | Mostly Dry |
| HQ | 2/20 | 1010 | 1156 | 67 | 302 | 20.2 | 1.39 | Melting |
| HQ | 2/26 | 1027 | 1157 | 66 | 309 | 20.4 | 1.29 | Melting |
| HQ | 3/04 | NA | 1105 | 57 | 342 | 19.5 | 1.56 | Melting |
| BA | 2/12 | 0946 | 1337 | 82 | 290 | 23.8 | NA | NA |
| BA | 2/20 | 1010 | 1224 | 80 | 290 | 23.2 | NA | Dry |
| BA | 2/26 | 1027 | 1139 | 82 | 290 | 23.8 | NA | Dry |
| BA | 3/04 | NA | 1116 | 76 | 307 | 23.3 | NA | Dry |

### 2.3.4 GPR Survey

We used GPR to estimate SWE along a transect for ground-based validation near the HQ site (Marshall et al., 2005; Webb, 2020). GPR data were collected on 12, 20, and 26 February at the same time as the snow pit data collection (Table 3). A GPR pulse is an electromagnetic wave that travels through the snowpack and is reflected off interfaces between materials with different dielectric properties such as $\rho_s$, with the strongest reflection often from the snow-soil interface at L-band (Bradford et al., 2009; Holbrook et al., 2016; Webb, 2017). For this study, two-way travel time ($t_2$) of GPR waves through the snow was obtained along transects using a Mala Geoscience, Inc. ProEx control unit pulse GPR system with an 800 MHz shielded antenna. The antenna was fixed in place on a plastic sled towed behind the operator. A GPS antenna connected to the ProEx control unit registered location information every second.

Radar pulses were triggered on 0.05 s intervals using eight times stacking (i.e., eight signals collected per point and averaged). The average survey travel speed was ~0.5 m s$^{-1}$ resulting in ~40 returns per meter. The ReflexW 2D Software package (Sandmeier, 2022) was used for time-zero adjustment, removing low-frequency background energy (i.e., dewow), and correcting for signal attenuation through the snow. For further details of GPR processing methods applied to snow, see Bonnell et al. (2021), McGrath et al. (2019), and Webb et al. (2018). The radar reflection from the snow–soil interface was then selected at the first break prior to the first peak of the reflection. A topographic correction was performed by dividing $t_2$ by the cosine of the ground surface slope at that location.

The $\epsilon_s$ of snow is sensitive to $\rho_s$ and LWC (Bradford et al., 2009; Heilig et al., 2015; Webb et al., 2018), and is related to the velocity ($v$) of the radar wave through snow:

$$v = \frac{s}{\sqrt{\epsilon_s}} \qquad (3)$$

where $s$ is the speed of light in a vacuum (~0.3 m ns$^{-1}$). For this study, the $\epsilon_s$ was directly measured in snow pit observations using an A2 Photonics WISe instrument at 10 cm vertical increments for the entirety of the snow pit height. We then averaged all WISe $\epsilon_s$ pit observations as the bulk $\epsilon_s$ value (Table 3). The observed $\epsilon_s$ was then used to estimate a velocity to distribute snow depth estimates along GPR transects.

$$d_s = \frac{vt_2}{2} \qquad (4)$$

These depth estimates were then converted to SWE by multiplying snow depth by the pit-observed bulk $\rho_s$ for direct comparison to UAVSAR-derived $\Delta$SWE (described further in Section 2.6). When using this approach of GPR observations in combination with a pit-observed bulk $\rho_s$, we expect to observe SWE values within 5 % at the frequency used (Marshall et al., 2005). We calibrated our GPR $\Delta$SWE data to the observed $\Delta$SWE at the snow pit. To do this, we defined a bias as the observed mean $\Delta$SWE difference between all GPR observations within 20 m of the snow pit and the SWE measured at the pit. We then removed this bias from the entire GPR dataset to create a directly comparable dataset relative to the UAVSAR-derived $\Delta$SWE. This is a similar method as described below to tie the UAVSAR data to snow pit observations. The point-based GPR returns were rasterized to the 6 m UAVSAR resolution and only those pixels with 30 or more GPR point observations were retained.

## 2.4   InSAR Atmospheric Correction

While radar signals can penetrate a moist and cloudy atmosphere, variation in dielectric properties between wet and dry air can significantly affect the radar signal (Ferretti et al., 2001). While substantial research has been conducted for correcting both tropospheric (Yu et al., 2018) and ionospheric (Meyer, 2011) effects from satellite-based SAR, suborbital SAR is both less common and has different correction considerations due to the lower acquisition altitude and often shallower and more diverse observation geometries.

Tropospheric atmospheric delay effects can be divided into two parts — dry delay and wet delay. The dry delay is caused by variations in temperature and pressure and is often considered less significant than the wet delay for spaceborne applications (Zebker et al., 1997). Wet delay is caused by spatial (within swath) and temporal (between acquisitions) variations in atmospheric water vapor concentrations (Danklmayer et al., 2009). Two recent studies (Michaelides et al., 2021; Bekaert et al., 2018) developed unique approaches to correct UAVSAR atmospheric delay. However, these methods were not directly applicable to the type of delay seen in our UAVSAR data.

As seen in Figure 6a, the 12–19 February uncorrected unwrapped phase data shows a noticeable near-to-far range phase ramp. For this UAVSAR flight, the average altitude was 12.9 km, compared to a satellite that traditionally orbits at ~750 km. This vastly lower sensing altitude causes a larger diversity of look angles and radar look vector length variations between the near and far ranges of the radar swath to emerge. The radar slant range, or the distance between a point on the ground and the

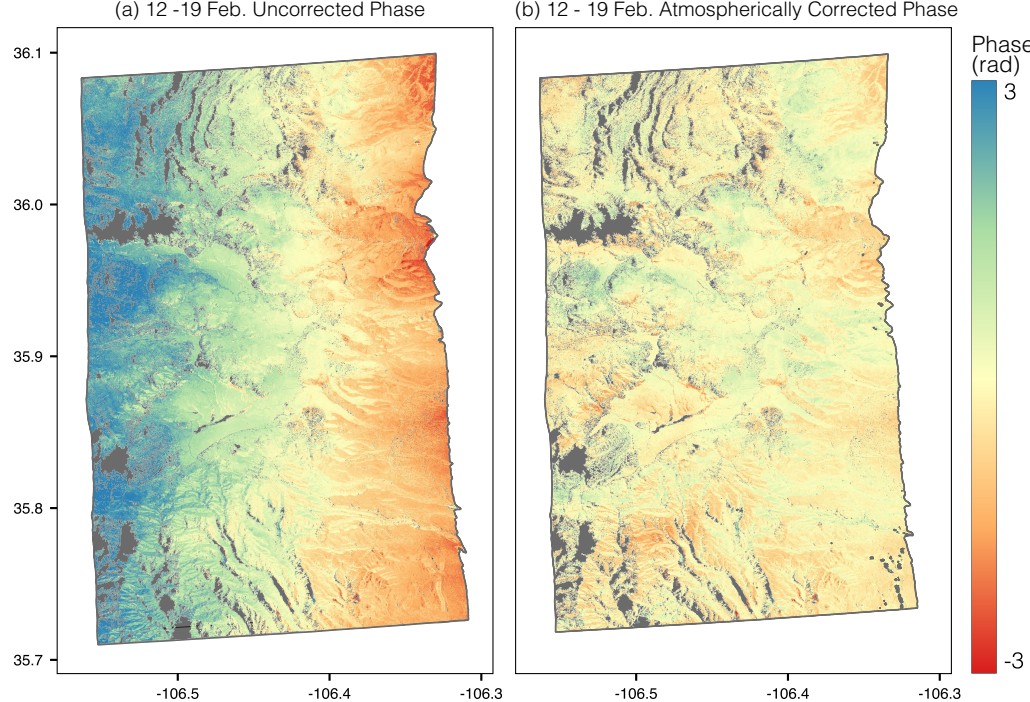

**Figure 6.** (a) The uncorrected phase from the 12–19 February pair and (b) the atmospherically corrected phase data. There is a linearly increasing phase ramp from near-to-far range (east to west), which is a distance of ~22 km.

radar, spanned from 11.4–27.8 km. The look angle varied from $28.51°$ in the near range to $69.01°$ in the far range. Thus, the radar wave in the far range is traveling through more atmosphere than the near range by a factor of 2.4.

Assuming a spatially homogeneous change in atmosphere between acquisitions, we used the pixel-wise slant range distance, or radar look vector (LKV), to correct the phase ramp. LKV is mostly dependent on the near-to-far range position in the scene but also varies with local topography. LKV is calculated by geocoding the east ($e$), north ($n$), and up ($u$) components of the single look complex (SLC) .lkv file (Supplement). The distance is then calculated by summing these components in quadrature:

$$LKV = \sqrt{e^2 + n^2 + u^2} \tag{5}$$

Phase values can be impacted by atmospheric delay and snowpack fluctuations simultaneously. By calculating the atmospheric delay of only snow free pixels defined by the 18 February fSCA product for the whole UAVSAR swath and comparing it to the atmospheric delay of only snow covered pixels, we were able to confirm the bulk of the observed signal is atmospheric and not snowpack related (Figure 7b). Using data from meteorological stations, we know there was not a large-scale snowfall event between the two flights. Using the linear relationship ($r^2 = 0.81$) between LKV and phase shown in Figure 7a, we subtract

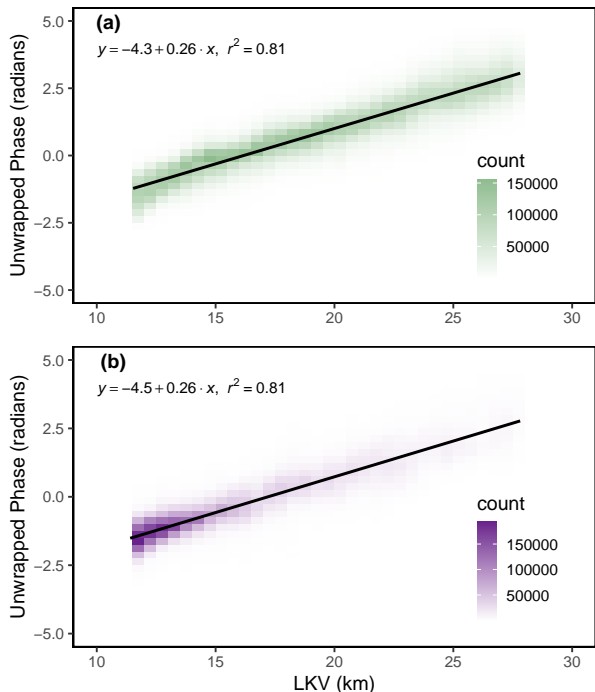

**Figure 7.** Density scatter plots showing the relationship between unwrapped phase and LKV in the 12–19 February InSAR pair for (a) snow covered pixels and (b) snow free pixels. The similarity in the two plots shows the large-scale phase signal is atmospheric and not snowpack related.

the estimated atmospheric component from the uncorrected data to achieve the atmospherically corrected image (Figure 6b). This correction method was applied to the 12–19 February and 12–26 February pairs.

### 2.5 Generating Local Incidence Angle Data

The local incidence angle ($\theta$) is the angle between the ground surface normal and the LKV on a per-pixel basis. The angle is calculated by deriving the surface normal from a DEM and computing the dot product with the LKV:

$$\theta = cos^{-1}(-\hat{n} \cdot LKV) \tag{6}$$

    Where $\hat{n}$ is the surface normal. $\theta$ varies with respect to local topography and the LKV. $\theta$ affects the distance the radar wave will travel through the snowpack and is a direct input into the SWE change inversion algorithm (Equation 2). We found errors

within the original SRTM DEM used in the UAVSAR data processing (Figure 8b). This error is likely due to phase noise in the SRTM interferograms as it is consistent throughout the dataset and falls within the known SRTM vertical uncertainty of $\pm$ 16 m (Rodríguez et al., 2006; Sun et al., 2003). VG is relatively flat and smooth outside of river channels and gullies. However, the original DEM shows artifacts on the order of 5–15 m throughout the meadow and does not accurately represent the ground

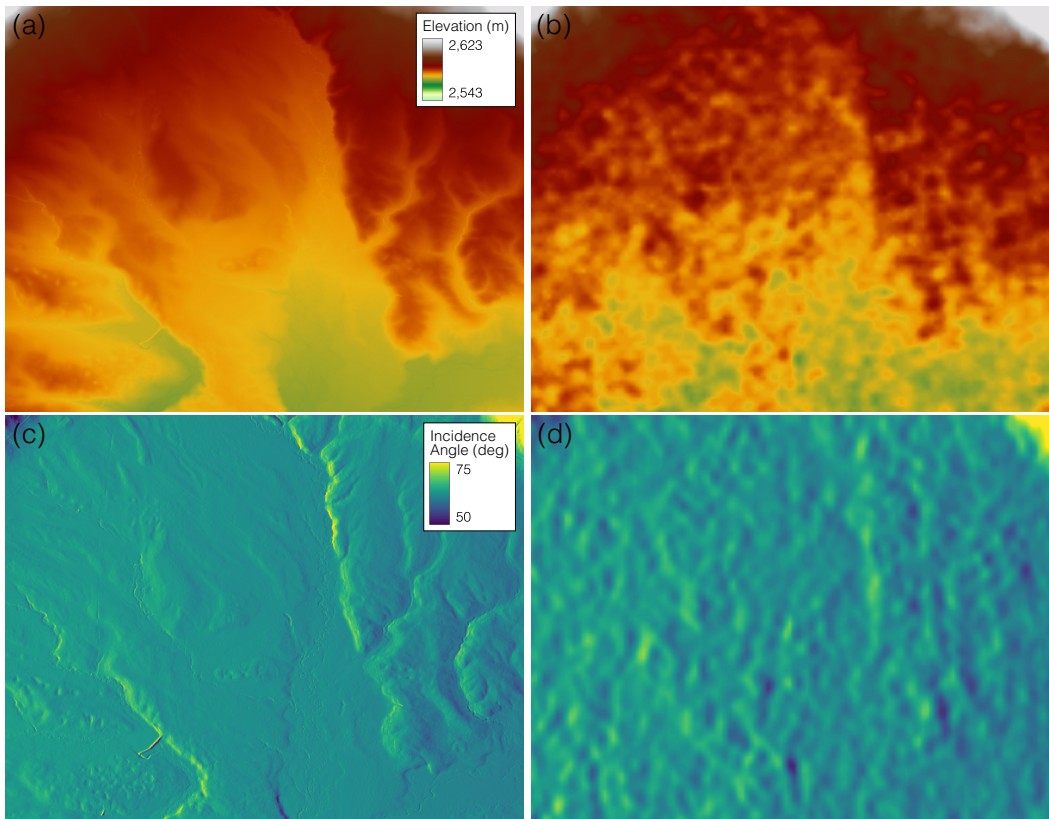

**Figure 8.** The snow free ground surface in a portion of the VG meadow for (a) the lidar DEM and (b) the SRTM DEM. UAVSAR $\theta$ is generated from (c) lidar and (d) the SRTM data. Gullies and small stream channels are easily discerned from the lidar DEM, while the SRTM DEM shows a variable surface with large mounds.

surface. These DEM artifacts propagate into the estimate of $\theta$ (Figure 8d), which is then input into the SWE change equation, causing errors in SWE change estimations.

We generated new $\theta$ data using a snow free lidar DEM (Figure 8a) acquired in 2010 for the Catalina-Jemez Critical Zone Observatory (CZO) (OpenTopography, 2012). The high spatial resolution of 1 m and elevation accuracy of 5–30 cm provides a more reliable starting point to calculate $\theta$. By using the LKV and the lidar DEM, the new $\theta$ data better represent the bare ground surface of VG (Figure 8c).

## 2.6 Calculating SWE Change

To begin the SWE change estimation, the three InSAR pairs were masked with Landsat fSCA data collected on 18 February 2020. All pixels with >15 % snow cover were included to not mistakenly exclude pixels in the forest where the snowpack is partially obstructed by forest canopy. Using Equation 2, $\Delta$SWE values were calculated on a pixel-wise basis with inputs of

$\lambda_i$ (23.84 cm), $\rho_s$, $\epsilon_s$, and the lidar-derived $\theta$. $\rho_s$ and $\epsilon_s$ are averages of the two snow pit values (Table 2) between the two
acquisition dates.

InSAR phase differences produce a relative measurement of change in SWE, therefore these data need to be tied to a point on the ground to estimate absolute change. Since there was near zero SWE change at the HQ snow pit between the three UAVSAR acquisitions (Table 3), we used this location as our InSAR known change point. This SWE change of −0.1 cm for 12–19 February and 0.2 cm for 19–26 February is well within the margin of measurement error (10 %) for the snow pit observations. To account for error within GPS snow pit location ΔSWE values for the snow pit pixel and the eight surrounding pixels were averaged. This averaged value was subtracted to obtain an absolute change. To calculate the cumulative ΔSWE, the two 7-day pairs were masked so only pixels that occurred in both scenes were considered and then added together.

## 3 Results

### 3.1 InSAR ΔSWE

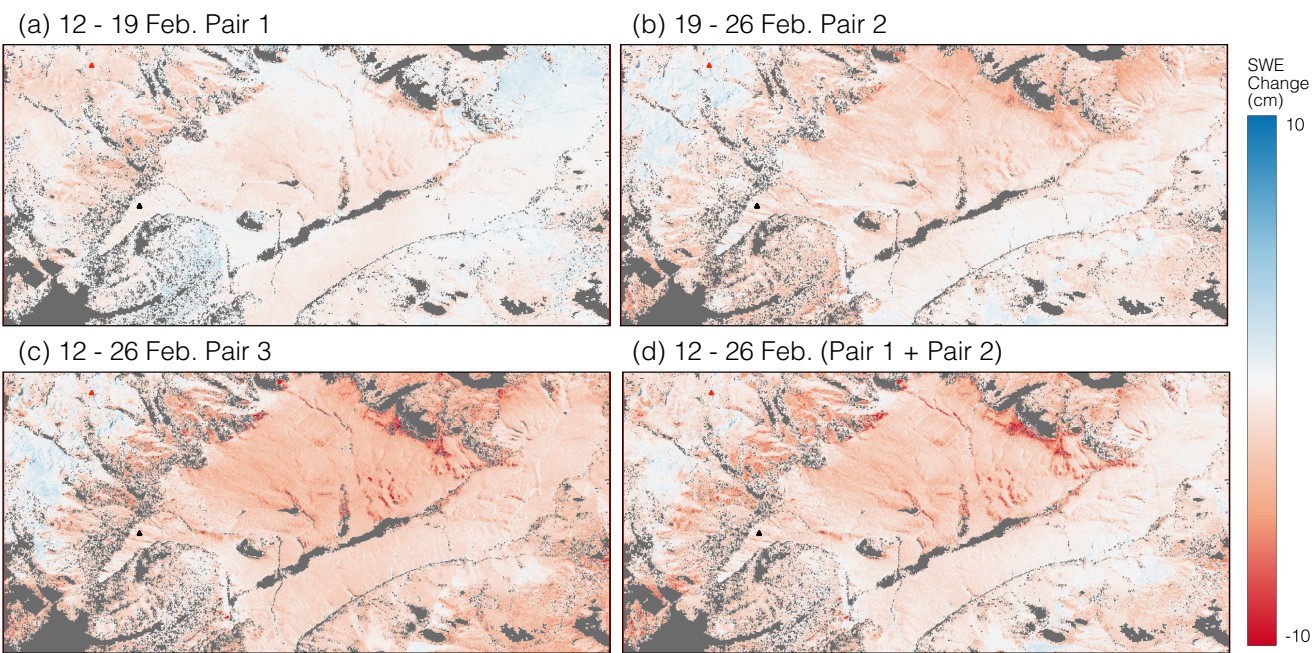

**Figure 9.** InSAR-derived ΔSWE results for (a) 12–19 February, (b) 19–26 February, (c) 12–26 February, (d) and the cumulative change between 12–26 February generated by adding the data from (a) and (b) together. The triangles represent the BA (red) and HQ (black) snow pits.

InSAR ΔSWE results are displayed in Figure 9 for (a) 12–19 February, (b) 19–26 February, (c) 12–26 February, and the (d) 12–26 February cumulative change (CM) in the study area. Table 4 reports ΔSWE mean, standard deviation (SD), median,

interquartile range (IQR), and is split into four physiographic classes. First is the full study area (Figure 2d), followed by the three defined in Figure 2f: VG, north facing slopes, and south facing slopes. Figure 10 shows histograms of InSAR-derived ΔSWE for the four aforementioned classes. We note that there was a greater mean estimated SWE loss for 19–26 February compared with 12–19 February for all physiographic regions (Figure 10a–d).

In Figure 9a (12–19 February), the full study area has a mean ΔSWE of −0.52 cm, with VG showing a similar change of −0.62 cm. In VG, the largest SWE losses occur in gullies and terrain depressions, with these areas showing visible SWE loss in all four pairs. The northeast corner of the study area shows a consistent increase in SWE, on the order of < 1 cm. There is a pattern of more SWE loss on the south facing slopes (mean = −0.58 cm) than the north facing (mean = −0.24 cm) for this pair.

Figure 9b (19–26 February) displays similar spatial patterns to those of Figure 9a. Overall the mean SWE loss was −1.24 cm, with the VG losing on average −1.34 cm. These SWE losses are over double that of 12–19 February. The highest elevation occurs in the northwest corner of the scene near Redondo Peak, and it is the only place to show consistent SWE increases. These increases are compared with in situ SWE data in the area (see Section 3.2). The pattern of more than double the SWE loss on south facing slopes (mean = −1.45 cm) compared to north facing (mean = −0.75 cm) continues from the first pair.

The 14-day baseline pair, 12–26 February (Figure 9c), has a mean ΔSWE of −2.29 cm. 12–26 February CM (Figure 9d), created by adding the values of Figure 9a and 9b together, has a mean value of −1.70 cm. While the IQR, SD, and histogram shape (Figure 10d–h) are similar in all four physiographic sections of the 14-day data, the mean of 12–26 February has a negative bias of ~0.5 cm compared to 12–26 February CM. This is likely due to variations in how these data were atmospherically corrected. The spatial patterns observed in the two 7-day InSAR pairs become amplified in both Figure 9c and 9d.

As a first-order estimate of uncertainty within the technique, we calculated the ΔSWE values for areas considered snow free by the 18 February fSCA data (Figure 4d). The ΔSWE data from the three pairs (12–19, 19–26, and 12–26 February) were combined, and we report a snow free ΔSWE mean value of −2.06 cm, an SD of 1.56 cm, and an IQR of 2.14 cm.

## 3.2 InSAR vs. Snow Depth Sensors, Snow Pits, and GPR ΔSWE

The InSAR-derived SWE retrievals were compared to three types of in situ SWE data: snow depth sensors, snow pits, and GPR. Figure 11a is a plot of ΔSWE values from the six CZO snow depth sensors and BA pit (~3030 m), and HQ Met snow depth sensor and pit (2650 m) against the InSAR ΔSWE values. Due to many of the in situ measurements being on or near the edge of a pixel, the InSAR ΔSWE values are an average of the pixel in which the measurement falls and the four closest pixels. The InSAR retrievals had a root mean square error (RMSE) of 1.46 cm and an MAE of 1.16 cm compared to the in situ measurements (n = 27, $r^2$ = 0.34). The small snowfall event noted in Section 2.3.3 is registered in the higher elevation CZO sensors and BA pit and not in the HQ Met location (Figure 5a). We see this same pattern for InSAR-based returns in Figure 9b (19–26 February), which is also shown by the mostly positive values of the pink dots in Figure 11a. The study area shows mostly SWE loss, while the higher elevation area in the northwest corner of the plot shows an increase indicating general agreement in the ablation and accumulation patterns.

We compared the InSAR and GPR ΔSWE between 12–26 February (Figure 11b). No significant relationship was found ($r^2$ = .042), and the RMSE and MAE increased to 3.03 cm and 2.57 cm, respectively. The error metrics were calculated using the

**Table 4.** ΔSWE (cm) mean, SD, median, and IQR from Figure 9 for the four InSAR pairs analyzed. They are split into the same four physiographic classes (full study area, VG, north facing slopes, and south facing slopes) as Figure 10. 12–26 February cumulative (CM) is created by adding the SWE changes from 12–19 February and 19–26 February pairs.

| ΔSWE (cm) | | | | |
| --- | --- | --- | --- | --- |
| Full Study Area | Mean | SD | Median | IQR |
| 12–19 Feb. | −0.52 | 1.11 | −0.50 | 1.18 |
| 19–26 Feb. | −1.24 | 1.30 | −1.12 | 1.64 |
| 12–26 Feb. | −2.29 | 1.68 | −2.24 | 1.87 |
| 12–26 Feb. CM | −1.70 | 1.54 | −1.53 | 1.86 |
| VG | | | | |
| 12–19 Feb. | −0.62 | 0.82 | −0.59 | 0.88 |
| 19–26 Feb. | −1.34 | 1.18 | −1.20 | 1.56 |
| 12–26 Feb. | −2.63 | 1.38 | −2.53 | 1.54 |
| 12–26 Feb. CM | −1.92 | 1.43 | −1.78 | 1.66 |
| North Facing Slopes | | | | |
| 12–19 Feb. | −0.24 | 0.98 | −0.23 | 1.04 |
| 19–26 Feb. | −0.75 | 1.26 | −0.62 | 1.45 |
| 12–26 Feb. | −1.46 | 1.51 | −1.58 | 1.78 |
| 12–26 Feb. CM | −0.97 | 1.27 | −0.83 | 1.47 |
| South Facing Slopes | | | | |
| 12–19 Feb. | −0.58 | 1.39 | −0.58 | 1.74 |
| 19–26 Feb. | −1.45 | 1.37 | −1.39 | 1.67 |
| 12–26 Feb. | −2.47 | 1.89 | −2.42 | 2.33 |
| 12–26 Feb. CM | −1.97 | 1.67 | −1.78 | 2.08 |

GPR data as validation, yet offsets in acquisition timing between UAVSAR and the GPR likely caused increased uncertainty when comparing the two datasets. On 12 February, the GPR acquisition began ~3 h after the 0946 LT UAVSAR flight, and on 26 February, the GPR data collection started ~2 h after the 1027 LT UAVSAR acquisition. During these acquisition time offsets, both temperature (Figure 5b) and incoming solar radiation (Figure 5d) were increasing. These atmospheric conditions presumably led to increases in snowpack LWC and $\epsilon_s$, which would explain why 44% of the GPR ΔSWE values showed increases when no measurable snowfall occurred in VG during the study period. We note that the presence of liquid water in the snowpack can cause a GPR signal delay that could be incorrectly interpreted as an increase in SWE. However, it should

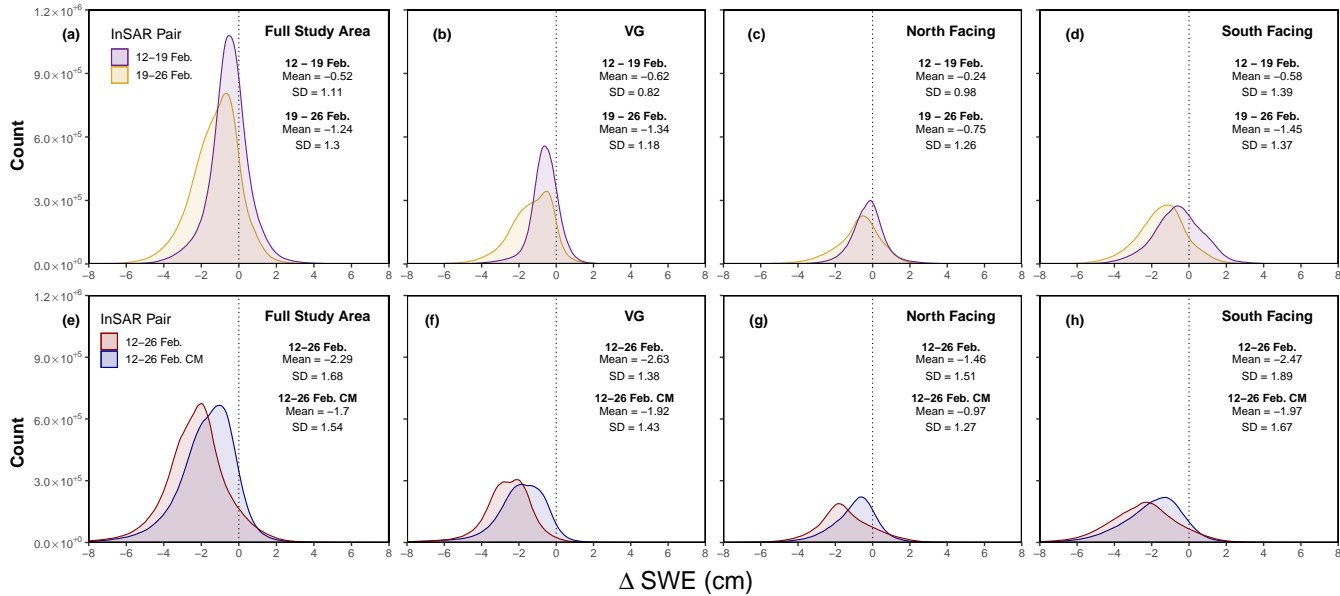

**Figure 10.** The distribution of ΔSWE values for the full study area, within VG, north facing, and south facing slopes. The top row displays the 12–19 and 12–26 February InSAR pairs (a–d), and the bottom row shows the 12–26 and 12–26 CM February InSAR pairs (d–h).

be stated that many of these points remain within the known uncertainty ($\pm$ 1 cm SWE) of L-band GPR observations for a dry snowpack (McGrath et al., 2019), with higher uncertainty expected during wet snow conditions. Furthermore, the mean
GPR derived ΔSWE product is ~0 cm, which matches well with the pit-observed change of ~0 cm (Table 3). The InSAR-derived ΔSWE product has a mean of −2.63 cm in VG; this indicates potential differences arise from using the pit observed $\epsilon_s$ measurements occurring later in the day than the InSAR retrievals and at the same time as the GPR survey. The potential change in snowpack properties that can occur during this time, as previously mentioned, could further explain these differences between the GPR and InSAR-derived products. However, it is important to note that these differences of 2–3 cm remain small
in the context of other remote sensing techniques, especially when considering complex spring snowmelt conditions.

### 3.3  ΔfSCA vs. InSAR ΔSWE

We compared the InSAR ΔSWE from 19–26 February (Figure 12a) to ΔfSCA between 18 February and 5 March (Figure 12b). The InSAR data were aggregated up to the 30 m Landsat resolution. While these datasets measure two different variables (SWE vs. fSCA) during different acquisition periods, the comparison of snow ablation (fSCA reductions) patterns provide
useful information when attempting to validate the experimental InSAR results. Several landscape features are prevalent in both datasets. The long gully that runs from the north central area of VG to the Jemez River is shown clearly in both maps. Other smaller gullies are also clearly visible. There are both SWE and fSCA losses on the south facing hillslopes surrounding the VG. Both of these patterns are being driven by these areas receiving more direct solar radiation. In the northwest corner of the image, the InSAR-derived map shows a small area of SWE increase, and the fSCA image shows no loss in this area.

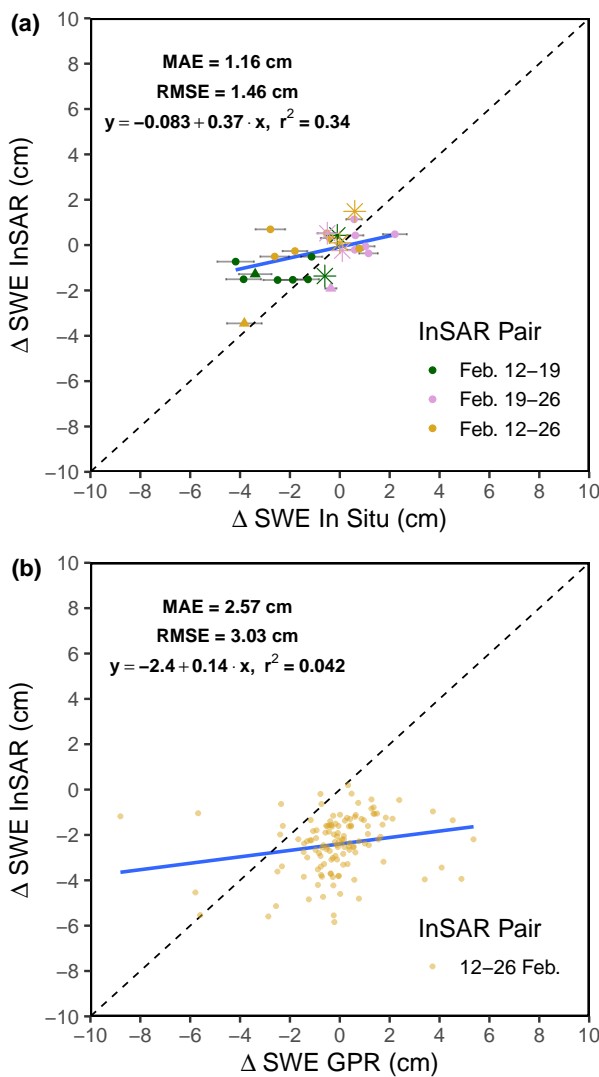

**Figure 11.** (a) Comparing in situ SWE changes from the six CZO sensors (circles), HQ snow depth sensor (triangles), and the BA and HQ pits (stars) to InSAR-derived SWE changes for the three InSAR pairs. The depth sensor SWE error bars are derived from a 10 % uncertainty from snow pit $\rho_s$ measurements and $\pm$ 1 cm uncertainty from the ultrasonic depth sensors. (b) Comparing InSAR and GPR derived $\Delta$SWE from 12–26 February.

Limited complete fSCA loss occurred in much of VG, while a mean value of −1.34 cm SWE was recorded. For optical images to show fSCA loss, bare ground must appear in the pixel. For the majority of the snow melt season, pixels lose SWE while still being completely snow covered. The fSCA product also shows more areas of melt than the $\Delta$SWE product, which can be attributed to the eight-day difference in the date of the last acquisition (26 February vs. 5 March). On 4 March, field teams reported widespread snowmelt throughout VG and the emergence of bare ground. The optical data also show areas of

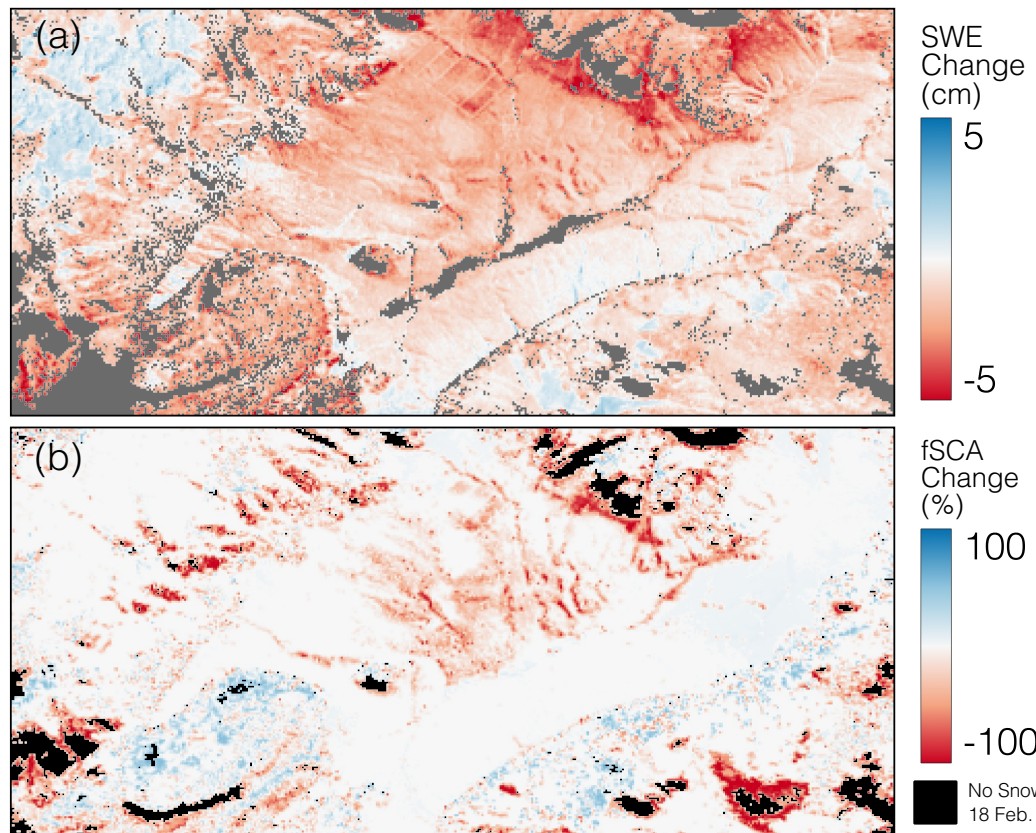

**Figure 12.** (a) InSAR ΔSWE between 19–26 February aggregated to the 30 m Landsat resolution, and (b) the Landsat ΔfSCA between 18 February and 5 March. The color scale for (a) was changed to −5 to 5 cm to exemplify the patterns.

100 % fSCA reduction and, therefore, bare ground appearing. fSCA gains are recorded in the densely forested hillslopes south of VG, which are shown by the true color imagery (Figure 4g and 4h). Uncertainty arises in forested areas from how the fSCA algorithm deals with sub-canopy snow estimation.

## 4  Discussion

### 4.1  Key Findings

During the study period, there was one localized measurable precipitation event on Redondo Peak, and temperatures were diurnally fluctuating below and above freezing (Figure 5b). With the snowpack going through daily partial freeze-thaw cycles, creating large sintered grains, and average wind speeds of 2.4 m s$^{-1}$ (Figure 5c), there is a low probability of blowing snow being a significant driver of SWE loss. Field team observations noted the hard surface of the snowpack during this time. This means that incoming solar radiation, causing surface melt and sublimation during the day, was the likely primary driver of

SWE loss. This is further confirmed by south facing slopes, which receive the most direct incoming solar radiation, showing about double the amount of SWE loss to that of north facing slopes for all InSAR pairs. These findings align with work by Musselman et al. (2008), who also observed midwinter SWE loss driven by incoming solar radiation in VCNP.

We hypothesize that the snowpack would become partially isothermal during the day, start to melt, and the surface would refreeze at night. The three UAVSAR flights occurred between 0930–1030 LT when the snowpack was still mostly frozen,
allowing the radar signal to hold coherence even though minimal LWC was still likely present in the snowpack. For SWE loss to occur with this hypothesis, melted snow needs to exit the snowpack or flow downslope. If melted snow is moving through the entire snowpack, it will not be entirely refrozen based on the meteorological data. It is possible that lateral flow within the snowpack (Webb et al., 2021a; Eiriksson et al., 2013; Evans et al., 2016) is moving snowmelt downslope between acquisitions.

Both the spatial distribution and magnitude of the ΔSWE patterns make sense, assuming insolation is the primary mechanism
driving SWE change during this time period. These patterns are confirmed by visually comparing the ΔSWE to ΔfSCA (Figure 12). There are noticeable similarities in the areas of greatest loss between the two datasets. The variation in acquisition time period and different parameters being measured do not allow for a direct quantitative comparison. However, when Marshall et al. (2021) quantitatively compared lidar snow depth changes to the UAVSAR phased-based depth retrievals, they found an $r^2$ of 0.76, an RMSE of 4.7 cm for snow depth, and 0.9 cm for SWE. These results add confidence to the findings presented
here and show similar RMSE values to the point-based snow depth sensor and snow pit comparisons presented in Figure 11a (RMSE 1.54 cm SWE).

The InSAR ΔSWE retrievals showed a stronger correlation to the snow pit and snow depth sensors ΔSWE compared to GPR. The depth sensors estimated SWE from snow height at a single point location and a bulk $\rho_s$ value from the nearby BA snow pit. GPR is a spatial observation that depends on the snow's dielectric properties, similar to InSAR retrievals. This makes
the radar methods for deriving SWE more sensitive to variability in snowpack properties such as density, LWC, and $\epsilon_s$. The GPR survey was conducted during mid-day when LWC can vary significantly as a result of increased solar radiation, which in turn increased the uncertainty in observations (e.g., 44 % of GPR pixels showed increasing SWE). The GPR measured some slight SWE increases, meaning there were increases in $\epsilon_s$; this is a sign that melt had begun during the afternoon acquisitions. Future GPR analyses will benefit from validation data collected over larger areas, synchronous timing with remote sensing,
and greater SWE variations between acquisitions. We believe GPR is a vital tool for future InSAR SWE validation efforts.

The 19–26 February pair is of particular interest because of the snowfall event (Figure 5a) that occurred on 22 February in the vicinity of Redondo Peak. This snow accumulation event was detected by the InSAR data, in situ snow depth sensors, and interval boards in the area of the BA snow pit. The lower elevations showed no accumulation in the InSAR retrievals, and this was confirmed by both the HQ met snow depth sensors and snow pit (Figure 11a). These results illustrate the ability to
track both snow ablation and accumulation within the same radar swath, furthering our confidence in the technique's ability to measure ΔSWE in a wide range of conditions. It is important to note that these small changes are within what can be an expected range of uncertainty for ΔSWE estimation due to LWC variations impacting the spatial variability of $\epsilon_s$ during spring snowmelt; capturing the spatial patterns within this range indicates great promise for future applications.

Leveraging morning acquisitions, we showed the UAVSAR L-band InSAR is able to maintain coherence over a 14-day baseline, even in the presence of diurnal melt cycles. The 12–26 February held coherence and provided quality snow phase information. This further supports the robustness of the technique for NISAR's 12-day repeating orbit. However, the biases between the 12–26 and 12–26 CM pairs, resulting from variations in their atmospheric correct, present additional complications. This is discussed in further detail in Section 4.2.

Corresponding research by Webb et al. (2021b) investigated the relationship between $\epsilon_s$ and LWC for both dry and wet snow conditions in the Jemez Mountains. With $\rho_s$ ranging between 261 to 309 kg m$^{-3}$ and $\epsilon_s$ between 1.26 to 1.39, snowpack LWC would range approximately between 3–5 %. This validates figures presented in Leinss et al. (2015), which state that at L-band (1.26 Ghz), and a $\rho_s$ of 300 kg m$^{-3}$, the radar signal can penetrate between about 10 m at 1 % LWC and 1 m at 10 % LWC. The high quality of the phase signal despite some snowpack LWC shows promise for the overall performance of NISAR and its 0600 and 1800 LT sun-synchronous orbit (Webb et al., 2021b; Bonnell et al., 2021).

## 4.2 Errors and Uncertainty

Our results provide an initial evaluation of InSAR-derived SWE uncertainty by reporting the mean (−2.06 cm) and SD (1.56 cm) ΔSWE values of snow free pixels. We note that these pixels only represent about 5 % of the study area, and much of the snow free area exists in densely forested regions where the fSCA uncertainty is greatest (Selkowitz et al., 2017). Section 4.3 outlines the continued work needed better to understand uncertainty within the InSAR SWE retrieval technique. In context with other SWE estimation techniques, airborne lidar has been shown to have uncertainty on the order of 7–8 cm for snow depth (Currier et al., 2019), and ~50 kg m$^{-3}$ for modeled $\rho_s$ (Raleigh and Small, 2017). This results in a similar magnitude of SWE uncertainty for the relatively shallow snowpack that develops in the VCNP.

The atmospheric correction developed in this study is specific to the UAVSAR data we used. It assumes a homogeneous delay related to the LKV. This delay is most likely due to pressure and temperature differences between radar acquisitions but does not account for smaller spatial scale water vapor variations in the atmospheric delay signal within the radar swath. While we're confident in the correction results from this method for the 12–19 and 12–26 February pairs, the consistency of near-to-far range phase ramp in these data is unique within the SnowEx UAVSAR dataset, and this method won't be directly applicable to all situations.

We held the $\epsilon_s$ values constant for the entire scene. While a single value may be sufficient for the VG meadow, the entire processed scene has more topographic and climatic variation, and therefore $\epsilon_s$ and $\rho_s$ variability within the snowpack. We used in situ measured $\epsilon_s$ values for this study to account for snowpack LWC, instead of estimating it from density like past studies (Rott et al., 2003; Deeb et al., 2011; Guneriussen et al., 2001). Eppler et al. (2022) and Leinss et al. (2015) attributed < ~5 % ΔSWE error to $\rho_s$ estimates. However, due to the known presence of LWC in the snowpack and the difference in timing between $\epsilon_s$ observations and UAVSAR flights, uncertainty is likely larger in our analysis. We showed that L-band InSAR could hold coherence with low (~1–5 %) levels of snowpack LWC. This adds complexity to the retrievals and should be the topic of future investigations. A variation in LWC between acquisitions will impact radar wave propagation speed and refraction angle in the snowpack, causing a phase shift that resembles a fluctuation in SWE, which could be either a gain or loss. The

ambiguity between LWC and SWE variations affecting $\phi_{snow}$ is resolved by using in situ data to understand the atmospheric and snowpack dynamics between the flights. For this reason, we limited the geographic scope of this study where field teams
evaluated snowpack conditions, motivated by our goal to confidently validate the ΔSWE retrievals.

The phase returns in this study were tied to a known change point using the in situ snow pit data. This method assumes that there was no variation in SWE or $\epsilon_s$ at this point between the three radar acquisitions. For future NISAR data, a time series could be initiated starting with a snow free scene. In such a scenario, any phase delay will be related to the new snow accumulated on the ground. The lack of temporal consistency of the suborbital UAVSAR measurements did not allow for the
implementation of this methodology.

We created new $\theta$ data using a high-resolution lidar DEM because of errors within the SRTM DEM. NISAR will use the TanDEM-X derived 30 m Copernicus DEM, which does not show the same inaccuracies as SRTM for non-vegetated areas (Rizzoli et al., 2017), and therefore will not be of significant concern. However, all further studies utilizing SnowEx UAVSAR data should inspect the $\theta$ raster provided before employing it in the SWE change inversion equation. If errors are found, new $\theta$
data should be generated using the Copernicus DEM or other methods (e.g., lidar) to minimize parameter uncertainty.

### 4.3   Future Work

The SnowEx 2020 and 2021 campaigns collected UAVSAR time series data at 14 different research sites across the WUS. While we reported a first-order estimate of uncertainty of ± 1.56 cm, future analysis of this large dataset should continue to quantify the uncertainties within the SWE retrieval technique. This includes but is not limited to: (1) the impacts of $\theta$, slope,
and aspect on the SWE returns; (2) considering the effect of snow wetness on Equation 2; (3) the influence various forest cover metrics; (4) constructing a consistent ΔSWE time series to prepare for NISAR's 12-day temporal repeat; and (5) implementation of spatially distributed $\rho_s$, $\epsilon_s$, and data into the SWE change equation. This could be derived from snowpack energy balance models (Marks et al., 1999; Liston and Elder, 2006) or through polarimetric radar retrievals (Shi and Dozier, 2000). Future NISAR InSAR SWE validation efforts would greatly benefit from synchronous airborne lidar snow depth acquisitions
with concurrent in situ measurements of $\epsilon_s$, $\rho_s$, and snow depth. These efforts should focus on complex mountain watersheds.

Previously, InSAR data has been used to measure geologic processes that vary at slower spatiotemporal scales than mountain SWE, and therefore image pairs could be selectively chosen to have minimal decorrelation and atmospheric effects. However, this is not the case for InSAR-based SWE monitoring; it requires a complete time series of snow accumulation and ablation throughout the winter season due to rapid decorrelation in snow covered regions.
The ability to confidently identify and correct for spatially and temporally varying atmospheric signals over mountain range scales is one of the main challenges facing this technique. To address this atmospheric limitation, additional orbital snow-specific correction methods must be developed. Future work should leverage past studies utilizing MODIS and other imaging spectrometers (Li et al., 2009), high-resolution weather models (Liu et al., 2009), GPS measurements (Li et al., 2006), and combinations of these techniques in tandem (Bekaert et al., 2015). While NISAR data products will include ionospheric and
tropospheric correction layers at 80 m spatial resolution, these corrections are automated and may not be temporally consistent enough for snow measurement purposes.

Furthermore, while the ΔSWE results are InSAR-derived, this technique requires a multisensor approach for correct implementation. Optical fSCA data are needed to identify snow covered pixels as part of the correction for atmospheric delay and to apply the SWE inversion equation over only snow covered pixels. The Landsat 8 image data used in this study represented two of the very few cloud-free days throughout the winter time series over the entire UAVSAR swath. To account for the significant issue of cloud cover, future investigations should leverage multiple optical sensors (e.g., Sentinel-2, MODIS, commercial high-resolution imagery, etc.), optical sensor fusion and interpolation methods (Rittger et al., 2021; Dozier et al., 2008), and focus on how to best combine SAR and optical data for SWE change monitoring. Any future SAR-derived SWE product such as the Ku- and X-band approach (Tsang et al., 2022), or the P-band Signals of Opportunity (SoOp) (Yueh et al., 2021) will require optical data to delineate snow covered pixels in midlatitude mountain environments, making this multisensor approach applicable for radars other than NISAR. Continued work on how to best fuse disparate sensors through cloud computing and machine learning will be key to progressing our knowledge of mountain snowpack monitoring (Durand et al., 2021).

## 5  Conclusions

This work leveraged high-resolution (6 m) UAVSAR interferometric data products to estimate ΔSWE at scales relevant to basin scale water resource management. We developed and applied a workflow utilizing UAVSAR data to detect both positive and negative changes in SWE. We then used in situ snow depth, $\rho_s$, ΔfSCA, and GPR data to validate the InSAR-based returns. These results show the robust ability of L-band InSAR to hold coherence and provide quality ΔSWE information even in relatively adverse conditions for radar remote sensing. This research is the first in a series of studies analyzing the SnowEx UAVSAR dataset in preparation for the launch of NISAR in early 2024.

NISAR's low latency (~2 days) cloud-based data products will provide the opportunity to implement this L-band InSAR SWE monitoring technique at continental scales. While there is significant progress needed to better understand uncertainties associated with the retrievals, NISAR's L-band InSAR will have the ability to estimate SWE in mountain regions globally. Spatiotemporally complete data will require a multisensor approach with optical data and assimilation into land surface models. We believe that NISAR has the potential to revolutionize the way SWE is measured from spaceborne remote sensing.

*Code and data availability.* The code and data used to perform this analysis and create the figures are publicly available at: https://doi.org/10.5281/zenodo. The uavsar_pytools package is archived at: https://doi.org/10.5281/zenodo.6578192. UAVSAR data are publicly available at the NASA Jet Propulsion Laboratory UAVSAR data portal (https://uavsar.jpl.nasa.gov/cgi-bin/data.pl, last access: 6 March 2023) and the Alaska Satellite Facility (ASF) Vertex data portal (https://search.asf.alaska.edu/). Landsat fSCA data (U.S. Geological Survey and Center, 2018) are publicly available through the United States Geologic Survey (USGS) EarthExplorer data portal (https://earthexplorer.usgs.gov/, last access: 6 March 2023). The Western Regional Climate Center (WRCC) climate station data are publicly available (https://wrcc.dri.edu/vallescaldera/, last access: 8 July 2022). SnowEx20 Jemez UNM 800 MHz MALA GPR, Version 1 data are publicly available at the NASA National Snow and Ice Data Center Distributed Active Archive Center (NSIDC) (https://nsidc.org/data/snowex, last access: 4 June 2022). Information on the SnowEx database: https://doi.org/10.5281/zenodo.7618107.

*Author contributions.* JT, RWW, and HPM conceptualized the overall study. JT and RWW performed the data processing and analysis. JT, RWW, and AWN drafted and edited the manuscript, with FJM and HPM providing helpful comments. AWN, HPM, and RWW provided financial support for the study.

*Competing interests.* The contact author declares that none of the authors have any competing interests.

*Acknowledgements.* Funding for this work was provided by the NASA grant numbers NNX17AL40G (PI: Nolin), 80NSSC18K1405 (Co-I: Webb), and NNX17AL61G (PI: Marshall). Bureau of Reclamation also provided funding with grant number R21AC10459 (Webb). We would like to thank Yunling Lou, Yang Zheng, the entire UAVSAR processing team; Dr. Noah Molotch, Leanne Lestak, and Dr. Adrian Harpold for providing snow depth data; and the SnowEx field teams, especially Adrian Marziliano, who collected in situ observations. The authors would also like to thank Zach Keskinen, Ross Palomaki, and Naheem Adebisi for their input and review of the code for this study. The authors are grateful to editor Alex Langlois and reviewers Cathleen Jones and Silvan Leinss for their thorough and insightful comments, which vastly improved the quality of this manuscript.

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
