# Peer review of "Estimating snow accumulation and ablation with L-band InSAR"

_The Cryosphere, 2022_

## Referee Comment (RC1)

**Review**
Estimating snow accumulation and ablation with L-band InSAR
by J. Tarricone et al.

**General comments**
This paper reports results of analysis of UAVSAR data acquired during the 2020
SnowEx campaign to evaluate the capability of L-band InSAR for measuring SWE.
Three acquisitions are used to form 3 interferograms, which are then compared to in
situ data.  The results are important as an early evaluation of the capability of L-band
InSAR for measuring SWE in dry or slightly wet snow and are particularly relevant
given the upcoming L-band NISAR mission.  This paper could be a significant
contribution to the literature, but needs some revisions.  I think that it will be even
more significant if more quantitative analysis is done including an estimation of the
uncertainty in the SWE derivation, and from that, recommendations for future
measurements.  Measurement of SWE is a priority identified in the 2017 Decadal
Survey and hence recommendations from the SnowEx campaign results could be are
needed to support the next Decadal Survey or NASA Explorer missions.

**Specific comments**
L3 - Here you say that the measurement of SWE is a challenge in mountain regions.  I suggest
the problem is more general and is a challenge with remote sensing, period.  Also, this first
sentence implies that your work is in a mountain region, but you specifically selected a site
without much topography to work in.  I'd change this from being so specific to something more
general.

L10 - I think of 'data fusion' as something more than what you did.  You did use both the optical
and SAR data, but not in a very sophisticated or novel way and not combining the information
together in an algorithm to get more information than available in either data set.  As far as I
can tell, the optical data was used to make a mask, then the InSAR applied to areas within the
mask.  Also, there was no analysis to show that the mask was necessary for the InSAR to work.  I
think of this as akin to using a land mask or other type of mask derived from optical imagery,
not real data fusion.  I think you are overstating the analysis.  I recommend 'novel method' or
even 'method' rather than 'novel data fusion method.'

L13-14 - This list of validation data sets corresponds to what you present in the paper.
Elsewhere you include fSCA in the list of validation data sets.  I'll point those locations out
below.

L63 - the phase is related to change in SWE = change in (density*depth), not to change in mass
directly.  It is a fine point indeed, but worth noting.

L97 - This is the first reference to a figure and it is called Figure 2. I prefer for the figures to be numbered in the order in which they are referred to in the paper. I don't know if Cryosphere requires that, but it is preferred.

L97 - You need to point to the workflow website here if it is open source. Also since this paper touts the workflow and code, it would be helpful to have an appendix or supplement describing the code in more detail. That doesn't go in the main body of the paper though.

Re. workflow website https://zenodo.org/record/7199836#.Y6Japi-B2eQ: I didn't find a README document in there describing what each python script does. Include that there and/or in an appendix to this paper.

L99 - Figure 2c is a lidar DEM not spatial change in SWE. The wording is confusing.

L100 - here you list fSCA as an evaluation product
L100 general - it would be helpful to the reader if you ended this paragraph by pointing out what will be discussed in the subsections.

L103 - "changes to the Earth's surface" - this statement is way too general.

L105-107 - Rosen citation goes after 'repeat pass InSAR' and you need to put individual citations after each topic listed. For example, Mouginot refers to ice sheets but is put after volcanic activity.

L108 - phase change related to CHANGE in dry snow SWE, not VARIATION is SWE.

L109 - The word 'rate' does not belong here. It refers to change over time. 'low attenuation' is sufficient and correct.

L119-120 - There are some issues here. 'Noise' is generally thought of as a random component or input from processing, e.g, sidelobes, but here you are lumping random and systematic errors together, but then ignoring the random noise in your description. Your biggest random noise comes from temporal decorrelation (Zebker et al., 1997). The biggest systematic uncertainty you are lumping in here is roughly as you describe, namely in knowledge of the plane's position. This should be its own term in the equation, but it doesn't appear often because for satellite InSAR the satellite position is much better determined. For UAVSAR that uncertainty is technically not with just the plane's GPS because the plane's position is determined using both GPS and an EGI. Yes, UAVSAR processing accounts for the plane's position as well as is known, but some phase change from uncompensated motion remains. That is the term that most impacts the UAVSAR phase.

L121 - 'phase influence from atmosphere' better described as 'phase contribution from change in path delay through the atmosphere'

L138 - total annual precipitation

Figure 1 and all figures - be consistent in use of (a), a, A, left, and so on in labeling the subpanels.

Fig. 1 Left - Take out the box.  It is in the text and frankly the font is way too small to read.  I don't like this figure at all and think it should be removed.  It doesn't add anything and the placement of phi_noise is wrong, per point made above.

Fig. 1 Right - The drawing is wrong.  Go back to Guneriussen to see why.  Del_Ra is not correct. You need to check whether that error propagated into your code.  (Just in case you need explanation to understand why the Guneriussen drawing is done as it is:  The drawing depicts incoming rays from an infinitely distant source that impinge on the same point on the ground. That is why the two lines depicting incoming rays are parallel to each other. )

L150 - Table 2 does not list which UAVSAR products were used in this study.  Somewhere you need to specify exactly which products you used.  Did you start from the SLCs, InSAR MLCs, InSAR GRDs, standard product = HH only, or quad-pol = special request?  Did you use the UAVSAR InSAR products in one case (to get their phase unwrapping) and UAVSAR SLCs in another, e.g., when you did your own processing?

L150-156 general - I have an issue with processing one pair one way and the others a different way.  Just process them all exactly the same way so that a one-to-one comparison can be made. In my opinion, this has to be done.  It is not optional.

Fig. 2-
1) (a) should be to the left of (b)
2) I don't see the value of (d)
3) Make (c) exactly the same extent as in Fig. 8.  I don't think that it is and that makes it hard to correlate the two.
4) Add slope map showing N- vs. S-facing slopes.  With the cut-off on (c)'s colorbar they are not all identifiable.
5) Show where the trees are.  This is an important point later in the paper but I can't tell where there are trees.
6) BA pit is not indicated by name in (c)

Fig. 3 -
1) Font is too small.
2) put colorbar outside the plot so that it can be better seen
3) Add map of delta fSCA to make change obvious.
4) you mention fSCA in VG meadow but I can't really see that.  Add a zoom image.

Table 1 -
1) bandwidth is 80 MHz

2) This table mixes technical specifications of the UAVSAR instrument with specs from the specific processed products used. I think that the last 6 refer to the products. Also, did you crop the near range to get a 16 km wide swath? UAVSAR scenes are generally 20-22 km wide. If you cropped it, why? Also, you mix specs for MLC products (az & slant rng spacing) with GRD products (ground range spacing). Did you georeference the MLCs yourself to that ground range spacing, in which case it isn't a UAVSAR spec. I can't tell exactly which products you used and what processing you did yourself.

L164 - Re 'stratigraphy' what exactly was measured? Are you saying that all the ones that follow were measured vs. depth?

Section 2.3.2 - provide uncertainties on the measured quantities. I think you mention some later in the text but that information belongs here.

L174 - You need to make it clear that this survey was done only near HQ. What days? times?

L176 - 'changes in material properties' is more correctly described as ' interfaces between material with different dielectric properties'

L181 - Describe what eight times stacking means for the non-expert

L183 - What is meant by 'first break'?

L184 - What is 'dewow'?

L185 - Is ' first break prior to the first peak of the reflection' the 'zero time'? Explain each better.

Eq. 3 - Discuss the assumptions, like uniformity of epsilon. What is the uncertainty in epsilon? What does that translate into as uncertainty in snow depth (eq. 4)? Is that uncertainty propagated into the comparison with SAR-derived SWE?

L191 - observed -> measured

L200 - you use the word 'tether' and I'm not familiar with it in this context. But when I think about it, I realize 'tie' isn't really any better, just more familiar to me. Your choice!

L206-207 - Are you listing the Michaelides paper because you followed that method? If I understand correctly, you only applied a high pass filter in what you did. (If that isn't the case, then a better explanation of the method you used is needed when you present it in later section.) If you are just presenting papers that corrected for atmosphere from airborne SAR, then Bekaert et al. can be included and their method is different than Michaelides'. There might be others that I am not familiar with.

Bekaert, D. S. P., C. E. Jones, K. An, M.-H. Huang (2018). Exploiting UAVSAR
for a comprehensive analysis of subsidence in the Sacramento Delta,
Remote Sensing of Environment, 220, 124-134,
doi:10.1016/j.rse.2018.10.023.

Paragraph around L220 - Why are you calculating this (PLV)? The UAVSAR SLC product contains the .lkv file, which gives the slant range in ENU components including accounting for the DEM = local topography. You can sum them in quadrature to get the slant range.
From UAVSAR product spec: *LKV file (.lkv): look vector at the target pointing from the aircraft to the ground, in ENU (east, north, up) components.*

L224 - Your equation 1 has snow as a separate term from atmosphere, not 'embedded' in it.

L224 - 226 - Re. ' By only calculating the atmospheric delay of snow free pixels from the Landsat fSCA product on 18 February, we were able to confirm...' - This isn't what you did. You compare the snow free pixels to the snow-on pixels, so you calculated it for both. Maybe you mean to say ' By calculating the atmospheric delay of only snow-free pixels from the Landsat fSCA product on 18 February and comparing to the atmospheric delay of only snow-on pixels...'

L229 - Was the same correction applied to all scenes or was the same method applied to get the correction? I'd think the latter but this says the former.

L235 - Only n_hat is previously undefined. Also 'site' -> 'sight'

Fig. 6
1) You need some commas
2) 'undulating' is the wrong word, it implies motion.
3) Did you use the incidence angle provided by JPL? I think you said previously that you calculated it. Please check for consistency and be clear throughout about where you used one and where you used the other.

L241 - 'mounds and undulations' are better described as 'artifacts' in this case.

General comments on section 2.6 -
1) So far fSCA is used only for generating a mask of snow-on/off and all of the 'fusion' relates to this product so it needs to be clear what its value is. What does this mask look like? Exactly how is it used?
2) Later in the paper 'masked' pixels are attributed to phase unwrapping. So what is the impact of the fSCA mask? Does the later mask not include it at all? Is it needed at all?
3) What was SWE in the fSCA-masked areas? This is a measure of the uncertainty/error in the SWE extraction. It is a good parameter to calculate and report.

L258 - change in SWE, not SWE

L258-259 - Be more precise in your description, what does 'tether' mean in this context? Calibrating?  Validating?

L282 - Your in situ measurement uncertainties on all the data need to be reported in the earlier sections where you describe the measurements.  Reporting them in the results section for the first time happens a lot, so I'm not going to mention it any more, just check throughout.

L263-264 - 'eight surrounding pixels' - did you exclude the snow pit pixel itself?  I can understand why you might, but it should be made clear.

L263 - Geocoding of UAVSAR is not a problem - see Fore et al.  You average to reduce the RANDOM errors, i.e., temporal decorrelation.

> Fore, A. G., Chapman, B. D., Hawkins, B. P., Hensley, S., Jones, C. E., Michel, T. R., & Muellerschoen, R. J. (2015). UAVSAR Polarimetric Calibration. IEEE Transactions on Geoscience and Remote Sensing, 53(6), 3481-3491.

Section 3.1 - This entire section should be placed earlier in the paper, before or after discussing in situ data.

L275 - what does 'preserved in phase unwrapping process' mean precisely?

L276 - You need more description of the ISCE processing.  Did you multilook?  Filter? Coherence depends strongly on multilooking so that needs to be specified.  Things like this are why I stated that the processing has to be done the same for all interferograms, otherwise comparisons don't mean much.

L281-282 - Re. 'the spatial variability in LWC...causes...' - this is assumed, not known via measurement.  If you want to state this then justify it, maybe with a reference, and discuss other possible sources.

L283 - Re. 'in this riparian area' - Okay, this is why I want a map showing where the trees are and where they aren't.  I can't check the images to verify what you are saying.  Also, are you saying the LWC is varying only in the riparian area?  What is the connection?

L283-284 - These 'artifacts' are definitely not the product of UAVSAR processing.  They look to me like RFI (radio frequency interference) from external sources, e.g., FAA radars.  I checked the UAVSAR products that I think you used and see that for the 2/12-19 pair those features show up in the coherence and the interferogram, but not in the unwrapped phase product.   That is probably because additional spatial filtering is applied during UAVSAR standard phase unwrapping.  My guess is that the details of the processing implemented in ISCE vs. used by the UAVSAR group are different, which is why the streaks don't show up in the 12-26 pair's coherence.  Certainly, they can't all be due to the Feb. 19 acquisition since all the streaks in the

12-19 pair's coherence aren't in the 19-26 pair's coherence.  Yet again I say to use exactly the same processing for all pairs.

Table 3 -
1) Include the 12-26 pair in this table.
2) The law of time reversal invariance tells us that the HV and VH backscatter from the surface must be the same.  Differences reported in this table cannot relate to real differences in the surface scattering.  Any differences in measured values come from errors in calibration, instrument cross talk, etc. Therefore, when making PolSAR products for UAVSAR, the average of HV and VH is used for a generic 'HV' product, the cross-polarization normalized radar cross section.  You should do the same in your analysis.  Otherwise, remove the values from the table since you only use the HH polarization in the end.  The HV and VH products are provided separately for people working with very dark scenes who want to estimate the noise, and that is not the case here.

L288-289 - I don't agree with this statement.  Most of the area around the VG meadow doesn't seem bad.  It certainly phase unwrapped.  If you are pointing out something important, it is definitely not obvious so add a figure and maybe quantify the difference.

L293 - values shown are not just in VG

L298 - It would be valuable for you to quantify the difference between S and N facing slopes.

3.2 Changes in SWE, general - You don't show the change in SWE for the BA area, but that is where the snowfall occurred.  Please show that since you are using HQ as your reference.  Being able to measure the SWE at BA is very important to your study, and if it is inconclusive or just very different then that too is important for estimating the uncertainty.  Your study is the first of its kind and it is important to report both success and limitations so that future studies can improve on it.  This is an opportunity to discuss in the conclusions what improvements need to be made (more snow, more frequent measurements, etc.)

 L303 - 'small storm' - all information about this and snowfall need to go in the data section, much earlier.  This includes Fig. 14.

L311 - Title needs to be Changes in SWE: InSAR vs. GPR and snow depth sensors, and hence you should combine sections 3.3 and the LAST SENTENCE of 3.4

L320 - I do not understand why this is 'likely' to be the MAXIMUM error.  A 5% uncertainty in GPR can be positive or negative.  Also, you didn't propagate all errors to get a maximum.

Section 3.4 -
1) All but the last sentence belongs much earlier in the data section.
2) What about discussing the results?  The bias is much less than for GPR (Fig. 12).  Some discussion of why is needed.

L338 - Why is the comparison only done for 12-26 pair? Shouldn't it be to the 19-26 pair? That is much closer in date to the optical data's dates and the snow fall event happened after 2/19. Compare to the 19-26 pair.

L344 - The loss on S-facing slopes is not easy to see. Like I mentioned above, we need a map showing slope directions and, hopefully, some quantification of differences between S and N. Alternatively, you could show plots with just the values on the S slopes and just the values on the N slope.

L353 - Like I mentioned above, we need a map showing where the forested areas are. You could overlay an outline on figures if you want to highlight something.

Fig. 7 - add points to show location of BA and HQ sites on the LH plots.

Fig. 9 - Add a comparison of values inside VG vs. outside VG. Also make it clear exactly what area is covered by the histogram. I think it is the entire scene extent in Fig. 7 but it isn't clear.

Fig. 10 - Instead of having V and H axes, have the 1:1 axis at 45deg. Also the colors for the two data sets are too similar.

Fig. 11 - This goes much earlier, in the data section.

Fig. 12 - Your fit includes Feb 12-26 CM. That overcounts the short temporal baseline pairs relative to the independent Feb 12-26 pair. Remove the points for Feb 12-26 CM from the graph and recalculate the fit.

Fig. 13 - As said above, use 19-26 Feb instead. Maybe add a plot of delSWE vs. DelfSCA to show the correlation better.

Fig. 14 - Move to the data section. Extend plot out to the end of the fSCA data. Show fSCA (Landsat) acquisitions as vertical lines also. Add plots of wind speed and maybe direction since you mention that information in section 4.1.

L368-369 - Sentence 'While ...' belongs in the data section.

L373-374 - I think that your argument at change in LWC caused the InSAR decorrelation also supports the statement that water moved downslope.

L378 - Given the lack of snowfall, I think you could do a quantitative comparison, as I've suggested above.

L381 - 'remarkably well' - I would avoid qualitative statements like this. I think this is an overstatement and without more quantitative comparisons I would not draw this conclusion.

L382-383 - I think that all your GPR data did was show a bias of -2 cm wrt InSAR, and that number disagreed with the snow depth measurements. Rather than general, and arguable, statements, use the in situ data to quantify the uncertainty and then use that to recommend different, i.e., more, InSAR measurements and different, ie., possibly more, in situ measurements in the future. Remember that this is one of the first studies and you can use it to justify and lay out a plan for the future. It wouldn't be a bad idea to reference the targeted observables of the Decadal Survey and suggest missions for it, possibly in conjunction with NISAR.

L383 - I suggest adding the need for higher SWE during the measurements.

L392 - The calculation goes in earlier in the paper, but you should add a discussion of the conclusions re. LWC change here.

L401 - I don't agree that this is known to be the maximum bias. I don't think that was shown.

L405 - This is the first mention of 'lightly forested areas'. I don't think that this paper as written has really explored the difference between forested (heavy or light) vs. unforested in a quantitative way. In the event that you do, then discussion is justified.

L408 - could be water content also

L416 - statement about Eppler study goes in data section, then can be referred to here.

L431-432 - Tandem-X is X-band, so likely to measure canopy height. Why do you think that it will be better at getting the land surface than SRTM? I didn't read the entire reference but it would be worth stating quickly why if this statement is correct.

L454 - In fact, you showed that it WASN'T necessary to identify snow-covered pixels to do the atmospheric correction. Or at least that is my take-away from Fig. 5.

L454-455 - You never showed the difference between using a snow-on mask vs. not using a snow-on mask anywhere in this paper. I cannot therefore conclude that it is necessary. In fact, the great value in using that is in determining the uncertainty in your SWE measurement, which was not done. It should be.

L460 - Ditto above - need for this was not demonstrated to this reviewer. If you think that you showed it, then make it more obvious.

L468 - List delfSCA as a validation data set here.

Data and code availability - general - Specific product names for everything you used needs to be provided.  It is not enough to point to huge databases where someone searches and guesses what exactly you used.

**technical corrections**
The entire manuscript needs to be read over to identify and correct errors in grammar, spelling, and language usage/wording.  I only point out a few below, but there are lots of places that deserve more attention.
Everywhere - 'data' is plural, so data are, data were
L36 - 1970s
L71 - pairs from the Envisat...
L72 - lack of in situ
L76 - its
L104 - 'can calculate' needs to be 'can be used to calculate' or 'is related linearly to'
L 140 - have two 'the'
L 207 - correct, not corrected
L335-336 - 'pixels they're located' -> deltaSWE
L415 - in situ measured epsilon...

**references**
incomplete citations for Mouginot , Brucker, Sandmeier, Selkowitz
L580, L681 - why all caps?
L607 - check, something is wrong

---

## Referee Comment (RC2)

2020-02-16 Sentinel-2 L1C RGB=Band (8,4,3).jpg

**2020-02-26 Sentinel-2 L1C RGB=Band (8,4,3).jpg**

---

## Author Comment (AC1)

**March 20, 2023**

Estimating snow accumulation and ablation with L-band InSAR
By: J. Tarricone et al.

**Response to Reviewer #2: Silvan Leinss**

Reviewer comments are shown in black. Responses are in blue.

> We added numbers to comments and letters for those with multiple parts. Comments in this response document are referenced using a #. All new text added to the manuscript is italicized.

**General comments:**

The authors describe the evaluation and interpretation of L-Band repeat-pass radar interferometry. Their aim is to observe changes of the snow water equivalent during partially wet snow conditions.

SWE estimation by radar interferometry is a very promising technique that has been developed within the past 20 years with increasing success. The main problems of this technique are a quick loss of coherence, correction for atmospheric phase delays, existence of phase reference points, and - especially for wet snow - the uncertainty and variability of the permittivity and signal penetration through the snow pack.

The authors tackle some of these problems and show that, even for melting conditions, coherence is maintained at L-band for the 7 and 14 day repeat time of their three acquisitions. They correct large-scale atmospheric phase delays by referencing the InSAR phase to snow free areas in the scene which is useful if no topography-dependent atmospheric phase exists. The authors reference the local InSAR phase to 8-9 pixels around a local snow pit and assume the remaining phase originates from changes in SWE (accumulation or melt) rather than local km-scale atmospheric delays. It is not clear where the km-scale phase patterns originate from (SWE change, permittivity changes or atmosphere).

Unfortunately, during the 14 days of the InSAR observation period, the snow pit data showed hardly any change in SWE which makes it very difficult for the authors to find a statistically convincing relation between the InSAR phase and SWE changes. Therefore, the authors rely on interpretation of local patterns which they assume to be caused by snow melt. Sentinel-2 imagery, as suggested by Simon Gascoin https://doi.org/10.5194/tc-2022-224-CC1, supports the authors assumption and should be considered. See provided images in the supplements (Channel 8 provides some information about snow wetness).

In addition to the unfortunate meteorological conditions I have some concern because the authors use the dry-snow-SWE-to-InSAR equation (Guneriussen 2001, Leinss 2015) even though wet snow (melting condidions) are considered. This equation depends on the permittivity of snow which changes significantly (from 1.5 to at least 2.2) when the snow becomes wet at constant SWE. Therefore, (some of) the observed phase change might be due to increasing wetness rather than a change in SWE.

The paper is very well structured and good to read, but need to be improved by better focusing on the main results; in addition to addressing the above mentioned concerns which could require a major revision of the paper.

> We would like to thank Dr. Leinss for his thoughtful comments, especially surrounding the topic of snow wetness and permittivity.
>
> While we've clarified the manuscript through the specific comments below, we wanted to make clear that our permittivity measurements were taken directly, not derived from snow density.
>
> We also note that Reviewer #1 recommended significant changes to the manuscript. Please refer to our replies there to see the full scope of the updates we have made.

**Specific comments:**

1. line 50: "where shorter wavelengths (..) have been used to estimate SWE (references)": The authors detail some technical challenges faced when estimating SWE from backscatter. It would be interesting, to provide a rough estimate or precision based on the conclusions from the cited authors, that apply these techniques, to indicate how well they were able to actually estimate SWE from backscatter. For example, accoording to the scattering physics that happen in snow, radar backscatter does not necessarily show a monotonically increasing relation to backscatter. I would claim, that determining the required in-situ parameters to derrive SWE might be even more difficult that determining SWE directly.

   > We agree that determining the in situ parameters for the volume scattering approach is exceedingly difficult, especially over heterogeneous terrain. The various implementations of this approach (theoretical, tower-based, airborne), and the different modeling frameworks implemented, make providing even a rough estimate of precision for the technique in general unfeasible.

2. L58: If the authors think it would be beneficial, they could cite Stefko/Leinss(2022) which provides a completely new approach to analyze the radar backscatter from snow (even though it does not claim to derrive SWE or snow height).

Stefko and Leinss et al. "Coherent backscatter enhancement in bistatic Ku- and X-band radar observations of dry snow", TC 2022, https://tc.copernicus.org/articles/16/2859/2022/

*While Stefko & Leinss (2022) is a fantastic paper, we don't think it fits into the scope of Section 1.2, which is focused on snow depth and SWE estimation techniques.*

3. L91: "accumulation and ablation": It is not totally clear what the difference is of this study compared to the study discussed in the paragraph before (Marshall 2021). In the previous paragraph "a wide range of (..) snow conditions" is specified. This study/paragraphs seems to adress "both snow accumulation and ablation". Do you mean ablation of dry snow by wind drift or evaporation, or do you mean ablation my melt an runoff? In line 92 you mention that the "UAVSAR-based approach" has been only applied to dry snow "but not melt". Does that indicate that you adress melt or, at least, wet snow periods? Try to better describe the differences betwen the two studies (you could also add the study side here, instead of in line 96.).

*In this context, the word 'ablation' was used to describe melt, evaporation, or sublimation. Grand Mesa is a high elevation (~3,400 m) cold location. During the study period of Marshall et al. (2021), there was no measurable precipitation, and the snow depth variations were driven by wind redistribution.*

*We updated the text to: "The overall goal of this study is to assess the performance of L-band InSAR for monitoring SWE changes in an environment where there is both snow accumulation and ablation (melt, evaporation, or sublimation). Currently, this UAVSAR-based approach has only been applied to cold dry snow conditions on Grand Mesa (Marshall et al. 2021), where the snow depth variations were mainly driven by wind redistribution, but not melt or evaporation. Towards this end, the specific objectives of the work presented here are to (1) analyze InSAR SWE retrievals over a complex mountain region, and (2) validate the retrievals using satellite and in situ data.."*

4. L25 - 130: The equation from Guneriussen requires knowledge about snow density rho_s and the snow permittivity epsilon_s. Due to a lack of spatially (and vertically) distributed information, these two variables seem to be determined from measurements in two snow pit data as written in line 255-257 which could potentially introduce a significant bias on the derrived SWE values. However, as shown in Leinss et al. (1015) [Figure 8left, Figure 9], for dry snow (and only for dry snow), there is an almost linear relationship between SWE and the InSAR phase which does hardly depend on snow density rho_s or epsilon_s. I think it is worth considering or mentioning this as it simplifies SWE determination.

Reference: Leinss et al. "Snow water equivalent of dry snow measured by differential interferometry", IEEE JSTARS (2015), https://doi.org/10.1109/JSTARS.2015.2432031

We determined epsilon_s in each snowpit vertically (10 cm segments) using the A2 Photonics WISe sensor.

This text has been clarified in Section 2.3.3: *"Measurements of snow depth, snow layer stratigraphy (grain size, grain shape, hand hardness, and manual wetness), $\rho_s$, $\epsilon_s$, and temperature were recorded for each pit. $\rho_s$, $\epsilon_s$, and temperature were measured in 10 cm increments starting at the top of the pit. Stratigraphic layer size is variable and defined by the observer. In situ $\rho_s$ measurements have been shown to have an uncertainty of ~10 \% (Conger and McClung, 2009; Proksch et al., 2016). $\epsilon_s$ was measured using an A2 Photonics WISe instrument (A2P, 2021), which Webb et al. (2021b) showed to have a mean absolute error (MAE) of 0.106 when compared to other in situ observations... Ultrasonic snow depth sensors have a known uncertainty of $\pm$ 1 cm (Ryan et al., 2008)."*

We also added: *"For dry snow, there is a direct relationship between $\epsilon_s$ and $\rho_s$, whereas for wet snow, the relationship becomes more complex, with even small amounts of liquid water vastly increasing $\epsilon_s$ values. Recent studies from Eppler et al. (2022) and Leinss et al. (2015) found that error in density estimates only biases total SWE change by $< \sim 5$ \% for dry snow in a wide range of $\theta$ (< 50$^\circ$) and $\rho_s$ (< 500 kg m $^{-3}$). Leinss et al. (2015) also showed a nearly linear relationship between $\Delta$SWE and interferometric phase for dry snow, which simplifies the SWE estimation. That said, we used Equation 2 because our study considers melting snow and $\epsilon_s$ is a direct input."*

5. L25 - 130, continued: In contrast to dry snow, for melting conditions, the linear relation between the InSAR phase change and SWE does not hold anymore. Please provide some information about the permittivity of wet snow compared to dry snow. From the references [Sihvola 1986, Webb 2021, Hallikainen 1986, fig. 9] I obtain that the real part of the permittivity increases from 1.5 to about 2.2...2.6 for the same parameters of density (300 kg/m^3), LWC (5%) and frequency (1 GHz). This should cause a significant change in the observed phase change even for a constant SWE.

For references, see comment about Table 2 further below.

Furthermore, the penetration depth into wet snow could vary considerably from the penetration into dry snow, at least for higher frequencies. For L-band and wet snow, the expected penetration should be checked to be larger than the snow depth.

This is an excellent point. The reason that this method still works is that we are taking observations of permittivity in the pits rather than calculating permittivity from density and liquid water. These pits are also dug a few hours after the UAVSAR acquisition, and likely slightly overestimating epilson_s. This will certainly be difficult in situations where direct observations of permittivity are not available, but that is outside the scope of this study as it could certainly be a future research paper on its own.

See #4 for updated text.

Snow penetration depth was confirmed in L393 - 394 (of original submission).

6. L129/130: Please mention how strongly the liquid water content can affect the real and imaginary part of the permittivity of wet snow. The real part has a stronger effect and changes are considerable. See further references and hints in the comment below for Table 2.

   See #5 above for updated text.

7. L132-142: To better understand the study area, I highly suggest showing Sentinel-2 images as suggested by Simon Gascoin to illustrate the land-cover of the studied area. There are S2 images from 2020-02-06, 2020-02-16, 2020-02-21, 2020-02-26.

   We added Landsat true color images from 18 Feb. and March 5th in Figure 4d and 4e, respectively.

8. Figure 1: The length Delta R_a does not correspond to the illustrations in (Guneriussen 2001, Leins 2015) and to the underlying physics of equation (2).

   We removed the left side of Figure 1 per the recommendation of Reviewer #1 and inserted an adapted version of Figure 7 from Leinss et al. (2015).

9. Note that it might be better to reference the peer reviewed paper of Guneriussen rather than the IGARSS proceeding (here and other places, possibly cite both).

   T. Guneriussen, K. A. Høgda, H. Johnsen, and I. Lauknes, "InSAR for estimation of changes in snow water equivalent of dry snow," IEEE Trans. Geosci. Remote Sens., vol. 39, Art. no. 10, 2001, doi: http://dx.doi.org/10.1109/36.957273.

   We added the correct citation. Thank you for catching this.

10. L118-121 (145-155): How was the interferometric data processed?

We standardized the interferometric processing per the request of Reviewer #1 (all done by the UAVSAR team at JPL) and added more information on how the InSAR data was processed in Section 2.3.1.

*"UAVSAR is a fully polarimetric L-band radar deployed on a NASA Gulf Stream III aircraft, traditionally flown at ~13,700 m with a 22 km nominal swath width (Hensley et al., 2008; Rosen et al., 2006). Detailed technical specifications of the radar are provided at the top of Table 1. UAVSAR data were accessed using the Python package uavsar\_pytools (Keskinen & Tarricone, 2022). It uses the asf\_search API (https://github.com/asfadmin/Discovery-asf\_search) for easier downloading, formatting, and analysis of UAVSAR data. The flights used in this study occurred in the mornings of 12, 19, and 26 February 2020. The UAVSAR team at the NASA Jet Propulsion Laboratory (JPL) processed two 7-day (12--19 and 19--26 February) and one 14-day (12--26 February) ground projected (GRD) InSAR pairs. They were unwrapped using the Integrated Correlation and Unwrapping (ICU) algorithm (Goldstein et al., 1988). Processing parameters are outlined at the bottom of Table 1, and information about the specific products used is provided in the Supplementary Material. For the three flights used in this study, the flight track baseline was maintained within $< \pm$ 3 m, which is within the $< \pm$ 5 m requirement (Hensley et al., 2008)."*

**Table 1.** Technical Specifications of the UAVSAR L-band radar (top). InSAR processing and data parameters (bottom).

| Parameter | Value |
| --- | --- |
| Wavelength | 23.84 cm |
| Frequency | 1.26 GHz |
| Polarization | Quad Pol |
| Bandwidth | 80 MHz |
| Pulse Length | 40 $\mu$s |
| Radar Look Direction | Left |
| Range Swath Width | 22 km |
| Average Near Range Look Angle | 28.01° |
| Average Far Range Look Angle | 68.9° |
| Ground Range Pixel Spacing | 6 m |
| Number of Looks in Range | 3 |
| Number of Looks in Azimuth | 12 |
| Phase Unwrapping Method | ICU |
| Phase Unwrapping Filtering Method | Low Pass |
| Phase Unwrapping Filter Window Size | 3 × 3 pixels |

Was there a perpendicular (across-track) baseline between the different flight tracks that caused a topographic phase that had to be removed?

There is no note of an across-track baseline for any of the three InSAR pairs used in our study. The UAVSAR team will leave a note on the data when there is an issue with the flight tracks.

Why was the SRTM used for processing and not a better DEM with lower height noise (see e.g. Fig. 6)?

SRTM is the standard DEM used in processing by the UAVSAR team. We're not exactly the reason for this. NISAR will utilize the Copernicus 30 m for InSAR processing.

Could the phase differences along the south-west exposed slopes be an artifact resulting from a DEM not perfectly coregistered with the radar data? Nevertheless, Sentinel-2 data suggest, that indeed, snow melt is observed here.

While this is a great thought, we don't believe the uncertainty with the SRTM DEM propagated into the geolocation of phase data. Below is a zoom in the unwrapped phase (left) from 12-19 Feb. compared to the lidar-derived north (blue)/south (orange) aspect classification right. The phase data aligns well with the aspect classification, with clear differences between the north/south facing slopes. Per the request of Reviewer #1, we added a quantitative comparison. See #62 in response to Reviewer #1.

[Figure]

11. Figure 2, caption: Are the six CZO snow depth sensors all at the same location? I see only one black diamond.

Captain has been clarified. The sensors are all relatively close together (~20 m), so they overlap each other at the scale of the map.

[Figure]

*"**Figure 2.** (a) DEM of the UAVSAR acquisition provided by NASA, with a red rectangle outlining the study area. (b) Map showing the area of the UAVSAR acquisition (black outline) in the Jemez Mountains, NM. (c) A close-up of the GPR transect outlined by the black rectangle in (d), with the HQ Met (blue triangle) and HQ snow pit (black triangle) displayed. Due to their close proximity, a single red triangle represents the BA pit and CZO snow depth sensors. Within the study area extent: (d) lidar DEM, (e) lidar-derived slope, (f) lidar aspect binned to north (270-90$^{\circ}$) (blue) and south (90-270$^{\circ}$) (orange) facing slopes, with the grey area representing the flat VG meadow where aspect values are not valid, and (g) NLCD canopy cover percentage of VG."*

12. L160: Section 2.3.2 (Snow Pit) contains no information about SWE estimation even though table 2 lists SWE values. How were the listed SWE values determined?

We added clarification about how SWE measurements were determined in Section 2.3.3 Snow Pit and Meteorologic Station Data and in the caption for Table 3.

Text updated to: *"Measurements of snow depth, snow layer stratigraphy (grain size, grain shape, hand hardness, and manual wetness), $\rho_s$, $\epsilon_s$, and temperature were recorded for each pit. $\rho_s$, $\epsilon_s$, and temperature were measured in 10 cm increments starting at the top of the pit. Stratigraphic layer size is variable and defined by the observer. In situ $\rho_s$ measurements have been shown to have an uncertainty of ~10 \% (Conger and McClung, 2009; Proksch et al., 2016). $\epsilon_s$ was measured using an A2 Photonics WISe instrument (A2P, 2021), which Webb et al. (2021b) showed to have a mean absolute error (MAE) of 0.106 when compared to other in situ observations… Ultrasonic snow depth sensors have a known uncertainty of $\pm$ 1 cm (Ryan et al., 2008)."*

13. L164/165 and table 2: Does the change of epsilon_s result from an increase in density or a change in liquid water content?

That is the observed epsilon_s values so it is from both increases in density and changes in liquid water. Given what we know about permittivity and snow properties, we could probably assume that much of those changes are probably from liquid water.

14. Table 2: The listed mean epsilon_s values appear to be only the real-value of the permittivity. However, as the snow condition is melting, I would at least mention the order of magnitude of the imaginary part of epsilon (permittivity of wet snow). For an imaginary part of epsilon on the same order (or larger) as the real part, equation (3) is not valid any more because the refractive index depends on both, the real and imaginary part. Fortunately, from references about the permittivity of wet snow around 1 GHz [Sihvola 1986, Webb 2021, Hallikainen 1986, fig. 9] I obtain eps"_wetsnow = 0.05..0.1 at 1 GHz, density 300 kg/m^3, liquid water volume content LWC=5%, which is by a factor of 5-10 smaller than the change of the real part of the permittivity: epsilon'_wetsnow increases from 1.5 to 2.2...2.6 for the same parameters of density, LWC and frequency.

Table 2 updated.

34. L235: "line of site" -> line of sight

Corrected.

35. Figure 11: Could you add to the caption of the figure the elevation of Redondo peak and HQ?

Added to the Figure 5 caption.

---

## Author Comment (AC2)

**March 20, 2023**

Estimating snow accumulation and ablation with L-band InSAR
By: J. Tarricone et al.

**Response to Community Commenter: Simon Gascoin**

Community comments are shown in black. Responses are in blue.

**Comment:**

Congratulations to the authors for this study. My comment is about this statement "Optical fSCA data are needed to identify snow covered pixels (..) The Landsat 8 image data used in this study represented two of the very few cloud free days throughout the winter time series over VCNP. To account for the significant issue of cloud cover, future investigations should leverage optical sensor fusion and interpolation methods (...)"

Instead of fusion and interpolation, Sentinel-2 provides Landsat-like data (high quality multispectral imagery at 10-20 m resolution) with a 5 day revisit time. Just in February 2020, I find 4 clear sky images over the study area in Valles Caldera National Preserve.

[Figure]

We would like to thank Dr. Gascoin for his commentary on our work. We choose to use Landsat 8 because the USGS produces an operational spectrally unmixed fSCA product. We have added the true color imagery from 18 February and March 5 to the updated fSCA plot below.

[Figure]

*"**Figure 4.** Landsat fSCA clipped to the UAVSAR swath extent (black outline) for (a) 18 February 2020 and (b) 5 March 2020. (c) The pixel-wise percent fSCA change between the two dates, with the black area representing 0 % fSCA from 18 February 2020. The study area (red box) (a) 18 February 2020, (b) 5 March 2020, (c) and the difference between the two dates. Landsat true color image in the study area for (d) 18 February 2020 and (e) 5 March 2020."*

*We also updated the language in the section you referenced, to "The Landsat 8 image data used in this study represented two of the very few cloud free days throughout the winter time series over the entire UAVSAR swath. To account for the significant issue of cloud cover, future investigations should leverage multiple optical sensors (e.g., t, MODIS, commercial high-resolution imagery, etc.), optical sensor fusion and interpolation methods ...."*

---

## Author Comment (AC3)

**March 20, 2023**

Estimating snow accumulation and ablation with L-band InSAR
By: J. Tarricone et al.

**Response to Reviewer #1: Cathleen Jones**

Reviewer comments are shown in black. Responses are in blue.

> We added numbers to comments and letters for those with multiple parts. Comments in this review document are referenced using a #. All new text added to the manuscript is italicized.

**General Comments**

This paper reports results of analysis of UAVSAR data acquired during the 2020 SnowEx campaign to evaluate the capability of L-band InSAR for measuring SWE. Three acquisitions are used to form 3 interferograms, which are then compared to in situ data. The results are important as an early evaluation of the capability of L-band InSAR for measuring SWE in dry or slightly wet snow and are particularly relevant given the upcoming L-band NISAR mission. This paper could be a significant contribution to the literature, but needs some revisions. I think that it will be even more significant if more quantitative analysis is done including an estimation of the uncertainty in the SWE derivation, and from that, recommendations for future measurements. Measurement of SWE is a priority identified in the 2017 Decadal Survey and hence recommendations from the SnowEx campaign results could be are needed to support the next Decadal Survey or NASA Explorer missions.

> We would like to thank Dr. Jones for her thorough review and thoughtful comments. We address all of the specific comments and summarize the major changes to the manuscript in the list below.
>
> 1. The 12-26 February data was reprocessed by the UAVSAR team. It was very similar to the ISCE processed pair and did not change the findings of the manuscript. These reprocessed data propagated through Figures (3, 9,10,11) and Table (2,4), which were subsequently updated.
>
> 2. We found an error in the geolocation of the CZO snow depth sensors. The location error of the group of sensors was approximately 500 m southwest of their correct location. Their correct location was determined using a 2013 UNAVCO terrestrial lidar scan of the sensors (personal comm., Dr. Adrian Harpold)

(https://tls.unavco.org/projects/U-032/PS01/SV01/). Dr. Noah Molotch confirmed that our revised geolocation was correct in the map below. The 500 m geolocation error did not significantly change the results shown in Figure 11 (In situ vs. InSAR SWE). In their revised locations, the snow depth sensors still captured the small snowfall event noted in the manuscript.

[Figure]

Rasterized 10 cm DEM and snow depth sensor locations derived from the UNAVCO terrestrial lidar scan of the area. The gray blobs are the actual tower locations, and the white lines are the tower arms detected in the cm-scale lidar scans.

[Figure]

Photo from near the BA snow pit location. A few of the snow depth sensors can be seen within the red circle.

3. We added to our InSAR SWE change analysis per the recommendation of many of your comments. To do that, we divided the study area into three physiographic classes: Valle Grande (relatively flat grassland), north facing slopes, south facing slopes (Figure 2f seen in #23). These classes are derived from the CZO bare-earth lidar DEM. The Results and Discussion sections have been updated to incorporate this new analysis. The updated text is included in the specific comments below.

4. We updated our interpretation of the GPR data in the Results and Discussion sections, adding more information on sources of uncertainty within the dataset. The updated text is included in specific comments.

5. As recommended in #95, we revised the grammar and language in the manuscript. This means certain sentences have slightly changed that aren't reflected in this document.

6. We added Supplementary Material with information on the specific data products we used. It is attached at the bottom of this document.

**Specific Comments**

1. L3 - Here you say that the measurement of SWE is a challenge in mountain regions. I suggest the problem is more general and is a challenge with remote sensing, period. Also, this first sentence implies that your work is in a mountain region, but you specifically selected a site without much topography to work in. I'd change this from being so specific to something more general.

   We removed "in mountain regions" to make the statement more general.

2. L10 - I think of 'data fusion' as something more than what you did. You did use both the optical and SAR data, but not in a very sophisticated or novel way and not combining the information together in an algorithm to get more information than available in either data set. As far as I can tell, the optical data was used to make a mask, then the InSAR applied to areas within the mask.

   Also, there was no analysis to show that the mask was necessary for the InSAR to work. I think of this as akin to using a land mask or other type of mask derived from optical imagery, not real data fusion. I think you are overstating the analysis. I recommend 'novel method' or even 'method' rather than 'novel data fusion method.'

We agree that "data fusion" may not be the correct terminology for our analysis.

Text updated to: *"We developed a snow-focused multisensor method that uses UAVSAR InSAR data synergistically with optical fractional snow covered area (fSCA) information."*

We address the comments about the necessity of the fSCA mask in #46-48.

3. L13-14 - This list of validation data sets corresponds to what you present in the paper. Elsewhere you include fSCA in the list of validation data sets. I'll point those locations out below.

   L10 updated from *"optical snow covered area (SCA)"* to *"optical fractional snow covered area (fSCA)"* and L13 from *"optical snow cover data"* to *"fSCA"*.

4. L63 - the phase is related to change in SWE = change in (density*depth), not to change in mass directly. It is a fine point indeed, but worth noting.

   We updated the text from "changes mass of dry snow" to *"SWE changes of dry snow"*.

5. L97 - This is the first reference to a figure and it is called Figure 2. I prefer for the figures to be numbered in the order in which they are referred to in the paper. I don't know if Cryosphere requires that, but it is preferred.

   We removed the reference to Figure 2 in L97, so they are now ordered correctly.

6. L97 - You need to point to the workflow website here if it is open source. Also since this paper touts the workflow and code, it would be helpful to have an appendix or supplement describing the code in more detail. That doesn't go in the main body of the paper though.

   We added a citation for the Zenodo repository of workflow.

   New text reads: *"First, we developed a workflow (Tarricone, 2023)..."*

7. Re. workflow website https://zenodo.org/record/7199836#.Y6Japi-B2eQ: I didn't find a README document in there describing what each python script does. Include that there and/or in an appendix to this paper.

   We added a README explain what each script does. We also added the data so each script will produce the given figure. Figures not noted in the workflow were made in QGIS with rasters produced from this workflow.

https://zenodo.org/record/7754560#.ZBkrsOzMKis

8. L99 - Figure 2c is a lidar DEM not spatial change in SWE. The wording is confusing.

   We removed the reference to Figure 2c.

9. L100 - here you list fSCA as an evaluation product

   This was updated for consistency, see #3.

10. L100 general - it would be helpful to the reader if you ended this paragraph by pointing out what will be discussed in the subsections.

    Text added: *"Section 2 (Methods) is split into the following subsections: 2.1 overviews InSAR for estimating SWE changes, 2.2 describes the study area, 2.3 reviews the remote sensing and in situ data, 2.4 is a description of the atmospheric correct steps, 2.5 explains the creation of new incidence angle data, and 2.7 outlines the SWE change calculation."*

11. L103 - "changes to the Earth's surface" - this statement is way too general.

    Text updated: *"InSAR is an active remote sensing technique that uses the differences in phase to map surface topography (single-pass) (Zebker and Goldstein, 1986)* **or various types of surface deformation** *(repeat-pass) (Goldstein and Zebker, 198y)"*

12. L105-107 - Rosen citation goes after 'repeat pass InSAR' and you need to put individual citations after each topic listed. For example, Mouginot refers to ice sheets but is put after volcanic activity.

    Rosen et al. (2000) moved to after 'repeat pass InSAR'. Funning et al. (2005) added for tectonic motion, Colesanti et al. (2003) for geomorphic processes, and Poland & Zebker (2022) for volcanic activity.

    Text updated: *"Traditionally repeat-pass InSAR (Rosen et al., 2000), where the sensor scans the same area at two different times, has been used to monitor tectonic motion (Funning et al., 2005), geomorphic processes (Colesanti et al., 2003), ice sheet velocity (Mouginot, 2012), and volcanic activity (Poland and Zebker, 2022)."*

13. L108 - phase change related to CHANGE in dry snow SWE, not VARIATION is SWE.

    Text updated to: *"For snow applications, Guneriussen et al. (2001) theorized a relationship between InSAR phase change and change in dry SWE between acquisitions."*

14. L109 - The word 'rate' does not belong here. It refers to change over time. 'low attenuation' is sufficient and correct.

    The word "rate" was removed.

    Text updated: *"Dry snow has a low attenuation of the radar signal..."*

15. L119-120 - There are some issues here. 'Noise' is generally thought of as a random component or input from processing, e.g, sidelobes, but here you are lumping random and systematic errors together, but then ignoring the random noise in your description. Your biggest random noise comes from temporal decorrelation (Zebker et al., 1997). The biggest systematic uncertainty you are lumping in here is roughly as you describe, namely in knowledge of the plane's position. This should be its own term in the equation, but it doesn't appear often because for satellite InSAR the satellite position is much better determined. For UAVSAR that uncertainty is technically not with just the plane's GPS because the plane's position is determined using both GPS and an EGI. Yes, UAVSAR processing accounts for the plane's position as well as is known, but some phase change from uncompensated motion remains. That is the term that most impacts the UAVSAR phase.

    We appreciate the clarification of the phi_noise term. We split this into two new variables; phi_random for random error and phi_systematic for systematic error.

    Text now reads: *"To isolate the SWE change impacts on the phase, other factors impacting phase must be identified and compensated for. Outlined in \cite{deebMonitoringSnowpackEvolution2011} and updated for suborbital acquisition considerations, total interferometric phase includes the following contributions:*

    *\begin{equation}*
    *\phi_{total} = \phi_{flat} + \phi_{topo} + \phi_{atm} + \phi_{snow} + \phi_{random} +\phi_{systematic}*
    *\end{equation}*

    *where $\phi_{flat}$ and $\phi_{topo}$ are phase impacts from flat Earth and local topography, which are both accounted for in the UAVSAR InSAR processing chain using the Shuttle Radar Topography Mission (SRTM) DEM as input. $\phi_{random}$ is the random error, where the majority comes from temporal decorrelation (Zebker et al., 1997). $\phi_{systematic}$ represents the systematic error within the UAVSAR instrument. This error is mainly associated with uncertainty in the plane's position and deviations in the flight track between acquisitions. Variations in the plane's position are*

*accounted for within the UAVSAR processing workflow as best as possible, but not all aircraft motion can be completely captured, which can leave residual phase change. \par*

*Assuming all previously mentioned errors are accounted for, extracting $\phi_{snow}$ from the observed phase $\phi_{total}$ in UAVSAR data mostly requires an accurate compensation of $\phi_{atm}$,* **which is the phase contribution from change in path delay through the atmosphere.** *Refer to Subsection 2.4 for a detailed explanation of how $\phi_{atm}$ is addressed in our approach"*

16. L121 - 'phase influence from atmosphere' better described as 'phase contribution from change in path delay through the atmosphere'

    See bolded in #15.

17. L138 - total annual precipitation

    Updated to: *"About 50% of the total annual precipitation…"*

18. Figure 1 and all figures - be consistent in use of (a), a, A, left, and so on in labeling the subpanels.

    We updated all the figures using the (letter) notation and marked them from left to right.

19. Fig. 1 Left - Take out the box. It is in the text and frankly the font is way too small to read. I don't like this figure at all and think it should be removed. It doesn't add anything and the placement of phi_noise is wrong, per point made above.

    The left side of Figure 1 has been removed.

20. Fig. 1 Right - The drawing is wrong. Go back to Guneriussen to see why. Del_Ra is not correct. You need to check whether that error propagated into your code. (Just in case you need explanation to understand why the Guneriussen drawing is done as it is: The drawing depicts incoming rays from an infinitely distant source that impinge on the same point on the ground. That is why the two lines depicting incoming rays are parallel to each other. )

    We updated Figure 1 to be an adaptation of the diagram from Lienss et al. (2015).

[Figure]

*"**Figure 1**. Diagram adapted from Leinss (et al. 2015) showing the geometric principle of the InSAR SWE retrieval. $R\_a$ represents propagation through atmosphere (no snow) and $R\_s$ with snow to the wave front. The amount of refraction ($\theta\_s$) and change in wave speed are controlled by $\epsilon\_s$, which is a function of snow $\rho\_s$. The variation in path length with and without snow to the wave front is equal to $\Delta R\_r$ - $\Delta R\_a$. This path length difference causes a phase delay which is used to estimate SWE changes."*

21. L150 - Table 2 does not list which UAVSAR products were used in this study. Somewhere you need to specify exactly which products you used. Did you start from the SLCs, InSAR MLCs, InSAR GRDs, standard product = HH only, or quad-pol = special request? Did you use the UAVSAR InSAR products in one case (to get their phase unwrapping) and UAVSAR SLCs in another, e.g., when you did your own processing?

We added Supplementary Material which details the extract products we used. See #25 for the updated Table 1.

New text: *"UAVSAR is a fully polarimetric L-band radar deployed on a NASA Gulf Stream III aircraft, traditionally flown at ~13,700 m with a 22 km nominal swath width (Hensley et al., 2008; Rosen et al., 2006). Detailed technical specifications of the radar are provided at the top of Table 1. UAVSAR data were accessed using the Python package uavsar\_pytools (Keskinen & Tarricone, 2022). It uses the asf\_search API (https://github.com/asfadmin/Discovery-asf\_search) for easier downloading, formatting, and analysis of UAVSAR data. The flights used in this study occurred in the mornings of 12, 19, and 26 February 2020. The UAVSAR team at the NASA Jet Propulsion Laboratory (JPL) processed two 7-day (12--19 and 19--26 February) and one 14-day (12--26 February) ground projected (GRD) InSAR pairs. They were unwrapped using the Integrated Correlation and Unwrapping (ICU) algorithm (Goldstein et al., 1988).*

*Processing parameters are outlined at the bottom of Table 1, and information about the specific products used is provided in the Supplementary Material. For the three flights used in this study, the flight track baseline was maintained within < $\pm$ 3 m, which is within the < $\pm$ 5 m requirement (Hensley et al., 2008)."*

22. L150-156 general - I have an issue with processing one pair one way and the others a different way. Just process them all exactly the same way so that a one-to-one comparison can be made. In my opinion, this has to be done. It is not optional.

    The UAVSAR team reprocessed the 12-26 February phase image pair for us. This means all image processing is standardized for all three pairs. The new data were incorporated into all subsequent analyses, figures, and corresponding text.

23. Fig. 2 -

    **1)** (a) should be to the left of (b)

    Changed.

    **2)** I don't see the value of (d)

    Panel (d) is the path taken by the GPR. It is provided to give spatial context to the extent of the data collection area. We feel it is necessary.

    **3)** Make (c) exactly the same extent as in Fig. 8. I don't think that it is and that makes it hard to correlate the two.

    Updated.

    **4)** Add slope map showing N- vs. S-facing slopes. With the cut-off on (c)'s colorbar they are not all identifiable.

    Added panel (f), which is a binned N vs. S slope map. We added a gray mask in the flat part of the VG meadow where aspect values are not identifiable.

    **5)** Show where the trees are. This is an important point later in the paper but I can't tell where there are trees

    NLCD canopy cover percent from 2016 (most recent) added in panel (g).

    **6)** BA pit is not indicated by name in (c)

    BA and HQ labeled added to panel (c).

Updated figure and caption:

[Figure]

*"**Figure 2.** (a) DEM of the UAVSAR acquisition provided by NASA, with a red rectangle outlining the study area. (b) Map showing the area of the UAVSAR acquisition (black outline) in the Jemez Mountains, NM. (c) A close-up of the GPR transect outlined by the black rectangle in (d), with the HQ Met (blue triangle) and HQ snow pit (black triangle) displayed. Due to their close proximity, a single red triangle represents the BA pit and CZO snow depth sensors. Within the study area extent: (d) lidar DEM, (e) lidar-derived slope, (f) lidar aspect binned to north (270-90$^{\circ}$) (blue) and south (90-270$^{\circ}$) (orange) facing slopes, with the grey area representing the flat VG meadow where aspect values are not valid, and (g) NLCD canopy cover percentage of VG."*

24. Fig. 3 -

**1)** Font is too small

Figure size increased.

**2)** put colorbar outside the plot so that it can be better seen

Color bar moved and enlarged.

**3)** Add map of delta fSCA to make change obvious.

Delta fSCA map added in (c).

**4)** you mention fSCA in VG meadow but I can't really see that. Add a zoom image.

Insert maps of the VG study area added for (d-f).

Updated figure and caption:

[Figure]

*"**Figure 4.** Landsat fSCA clipped to the UAVSAR swath extent (black outline) for (a) 18 February 2020 and (b) 5 March 2020. (c) The pixel-wise percent fSCA change between the two dates, with the black area representing 0 % fSCA from 18 February 2020. The study area (red box) (a) 18 February 2020, (b) 5 March 2020, (c) and the difference between the two dates. Landsat true color image in the study area for (d) 18 February 2020 and (e) 5 March 2020."*

25. Table 1 -

**1)** bandwidth is 80 MHz

Corrected.

**2)** This table mixes technical specifications of the UAVSAR instrument with specs from the specific processed products used. I think that the last 6 refer to the products. Also, did

you crop the near range to get a 16 km wide swath? UAVSAR scenes are generally 20-22 km wide. If you cropped it, why?

This information was pulled from Hensley et al. (2008). Upon further review, the actual width is 22 km. We have updated the information.

Also, you mix specs for MLC products (az & slant rng spacing) with GRD products (ground range spacing). Did you georeference the MLCs yourself to that ground range spacing, in which case it isn't a UAVSAR spec. I can't tell exactly which products you used and what processing you did yourself.

We split Table 1 into two sections. The top includes the technical specification of the UAVSAR, and the bottom is the InSAR processing parameters.

**Table 1.** Technical Specifications of the UAVSAR L-band radar (top). InSAR processing and data parameters (bottom).

| Parameter | Value |
|---|---|
| Wavelength | 23.84 cm |
| Frequency | 1.26 GHz |
| Polarization | Quad Pol |
| Bandwidth | 80 MHz |
| Pulse Length | 40 $\mu$s |
| Radar Look Direction | Left |
| Range Swath Width | 22 km |
| Average Near Range Look Angle | 28.01° |
| Average Far Range Look Angle | 68.9° |
| Ground Range Pixel Spacing | 6 m |
| Number of Looks in Range | 3 |
| Number of Looks in Azimuth | 12 |
| Phase Unwrapping Method | ICU |
| Phase Unwrapping Filtering Method | Low Pass |
| Phase Unwrapping Filter Window Size | 3 × 3 pixels |

26. L164 - Re 'stratigraphy' what exactly was measured? Are you saying that all the ones that follow were measured vs. depth?

The observer defines the stratigraphic snow layers. Then, on each layer, they record data on snow grain size, grain shape, hand hardness, and manual wetness.

Text updated to: *"Measurements of snow depth, snow layer stratigraphy (grain size, grain shape, hand hardness, and manual wetness), $\rho_s$, $\epsilon_s$, and temperature were recorded for each pit. $\rho_s$, $\epsilon_s$, and temperature were measured in 10 cm increments starting at the top of the pit. Stratigraphic layer size is variable and defined by the observer. In situ $\rho_s$ measurements have been shown to*

*have an uncertainty of ~10 \% (Conger and McClung, 2009; Proksch et al., 2016).*
*$\epsilon_s$ was measured using an A2 Photonics WISe instrument (A2P, 2021), which*
*Webb et al. (2021b) showed to have a mean absolute error (MAE) of 0.106 when*
*compared to other in situ observations… Ultrasonic snow depth sensors have a known*
*uncertainty of $\pm$ 1 cm (Ryan et al., 2008)."*

27. Section 2.3.2 - provide uncertainties on the measured quantities. I think you mention some later in the text but that information belongs here.

Uncertainty values for in situ snow density, permittivity, and snow depth have been added. See #26 for updated text.

28. L174 - You need to make it clear that this survey was done only near HQ. What days? times?

Text updated to: *"We used GPR to estimate SWE along a transect for ground-based validation near the HQ site (Marshall et al., 2005; Webb, 2020). GPR data were collected on 12, 20, and 26 February at the same time as the snow pit data collection (Table 3)."*

29. L176 - 'changes in material properties' is more correctly described as ' interfaces between material with different dielectric properties'

Text updated to: *"A GPR pulse is an electromagnetic wave that travels through the snowpack and is reflected off interfaces between materials with different dielectric properties such as $\rho_s$..."*

30. L181 - Describe what eight times stacking means for the non-expert

Text updated to: *"Radar pulses were triggered on 0.05 s intervals using eight times stacking (i.e., eight signals collected per point and averaged)."*

31. L183 - What is meant by 'first break'?

We have simplified the processing jargon for non-radar users. These methods are common for all GPR users, but not necessarily important to go into detail for the context of this paper.

Text updated to: *"The ReflexW 2D Software package (Sandmeier and Straße, 2022) was used for time-zero adjustment, removed low frequency background energy (i.e., dewow), and corrected for signal attenuation through the snow. For further details of GPR processing methods applied for snow, see Bonnell et al. (2021), McGrath et al. (2019), and Webb et al. (2018)."*

32. L184 - What is 'dewow'?

    See #31.

33. L185 - Is ' first break prior to the first peak of the reflection' the 'zero time'? Explain each better.

    See #31.

34. Eq. 3 - Discuss the assumptions, like uniformity of epsilon. What is the uncertainty in epsilon? What does that translate into as uncertainty in snow depth (eq. 4)? Is that uncertainty propagated into the comparison with SAR-derived SWE?

    L190 old text: "For this study, the ϵs was directly observed in snow pit observations using an A2 Photonics WISe instrument."

    Text updated to: *"For this study, the $\epsilon_s$ was directly measured in snow pit observations using an A2 Photonics WISe instrument at 10 cm vertical increments for the entirety of the snow pit height. We then averaged all WISe $\epsilon_s$ pit observations as the bulk $\epsilon_s$ value (Table 3)."*

35. L191 - observed -> measured

    See #34.

36. L200 - you use the word 'tether' and I'm not familiar with it in this context. But when I think about it, I realize 'tie' isn't really any better, just more familiar to me. Your choice!

    Tether updated to tie here and at all other instances throughout the manuscript.

37. L206-207 - Are you listing the Michaelides paper because you followed that method? If I understand correctly, you only applied a high pass filter in what you did. (If that isn't the case, then a better explanation of the method you used is needed when you present it in later section.) If you are just presenting papers that corrected for atmosphere from airborne SAR, then Bekaert et al. can be included and their method is different than Michaelides'. There might be others that I am not familiar with.

    Bekaert, D. S. P., C. E. Jones, K. An, M.-H. Huang (2018). Exploiting UAVSAR for a comprehensive analysis of subsidence in the Sacramento Delta, Remote Sensing of Environment, 220, 124-134, doi:10.1016/j.rse.2018.10.023.

    We were presenting other papers that developed atmospheric correction methods for UAVSAR.

Text now reads: *"Two recent studies (Michaelides et al., 2021; Bekaert et al., 2018) developed unique approaches to correct UAVSAR atmospheric delay. However, these methods were not directly applicable to the type of delay seen in our UAVSAR data."*

38. Paragraph around L220 - Why are you calculating this (PLV)? The UAVSAR SLC product contains the .lkv file, which gives the slant range in ENU components including accounting for the DEM = local topography. You can sum them in quadrature to get the slant range. From UAVSAR product spec: *LKV file (.lkv): look vector at the target pointing from the aircraft to the ground, in ENU (east, north, up) components.*

    This is exactly what we did. For clarity and consistency with the UAVSAR file naming conventions, PLV has been changed to LKV throughout the manuscript.

39. L224 - Your equation 1 has snow as a separate term from atmosphere, not 'embedded' in it.

    Text updated to: *"Phase values can be impacted by atmospheric delay and snowpack fluctuations simultaneously."*

40. L224 - 226 - Re. ' By only calculating the atmospheric delay of snow free pixels from the Landsat fSCA product on 18 February, we were able to confirm...' - This isn't what you did. You compare the snow-free pixels to the snow-on pixels, so you calculated it for both. Maybe you mean to say ' By calculating the atmospheric delay of only snow-free pixels from the Landsat fSCA product on 18 February and comparing to the atmospheric delay of only snow-on pixels...'

    This wording was incorrect. Thank you for catching this.

    Text updated to: *"By calculating the atmospheric delay of only snow free pixels defined by the 18 February fSCA product for the whole UAVSAR swath and comparing it to the atmospheric delay of only snow covered pixels…"*

41. L229 - Was the same correction applied to all scenes or was the same method applied to get the correction? I'd think the latter but this says the former.

    The same correction method was applied

    Text updated to: *"This correction method was applied to the 12--19 February and 12--26 February pairs."*

42. L235 - Only n_hat is previously undefined. Also 'site' -> 'sight'

*Definitions for theta_i and LVK have been removed.*

*Text updated to: "Where $\hat{n}$ is the surface normal."*

*Corrected: "...line of sight..."*

43. Fig. 6

**1)** You need some commas

*Commas added.*

**2)** 'undulating' is the wrong word, it implies motion.

*Fig. 6 caption updated to, "...while the SRTM DEM shows a variable surface with large mounds."*

**3)** Did you use the incidence angle provided by JPL? I think you said previously that you calculated it. Please check for consistency and be clear throughout about where you used one and where you used the other.

*We used the lidar-derived incidence angle in our analysis.*

*Stated in L259: " Using Equation 2, $\Delta$SWE values were calculated on a pixel-wise basis with inputs of $\lambda_i$ (23.84 cm), $\rho_s$, $\epsilon_s$, and the **lidar derived $\theta$.**"*

*New figure and caption:*

[Figure]

*"**Figure 8.** The snow free ground surface in a portion of the VG meadow for (a) the lidar DEM and (b) the SRTM DEM. UAVSAR $\theta$ generated from (c) lidar and (d) the SRTM data. Gullies and small stream channels are easily discerned from the lidar DEM, while the SRTM DEM shows a variable surface with large mounds."*

44. L241 - 'mounds and undulations' are better described as 'artifacts' in this case.

    Text updated to: *"However, the original DEM shows artifacts on the order of 5--15 m throughout the meadow and does not accurately represent the ground surface"*

General comments on section 2.6 -

45. 1) So far fSCA is used only for generating a mask of snow-on/off and all of the 'fusion' relates to this product so it needs to be clear what its value is. What does this mask look like? Exactly how is it used?

    Refer to #24 for the updated fSCA figure. We also created Section 2.3.2 'Landsat fSCA' to better describe the utility of this data.

    Text in that section now reads: *"No current technique can confidently discriminate dry snow cover using solely L-band radar (Tsai et al., 2019). Our study aims to assess the*

*ability of L-band InSAR to estimate spatiotemporal SWE changes. Therefore, our analysis requires properly identifying snow covered pixels within the UAVSAR swath, ensuring the radar signal interacts with mostly snow cover and not bare ground. To do this, we utilized Landsat 8 fSCA (U.S. Geological Survey and Center, 2018) data from 18 February and 5 March 2020 (Figure 4). These data are generated using a spectral unmixing analysis based Snow Covered Area and Grain size (SCAG) algorithm developed for MODIS (Painter et al., 2009). The data processing workflow includes water masking, cloud masking, and canopy cover corrections (Selkowitz et al., 2017; Stillinger et al., 2023). Within the full UAVSAR swath, 29.7 % of pixels were entirely snow-free on 18 February (Figure 4a), increasing to 38.1 % on 5 March (Figure 4b). For just the study area, 4.1 % pixels were snow free on 18 February (Figure 4d), with an increase to 9.1 % by 5 March (Figure 4e)."*

46. 2) Later in the paper 'masked' pixels are attributed to phase unwrapping. So what is the impact of the fSCA mask? Does the later mask not include it at all? Is it needed at all?

Pixels are lost in the phase unwrapping processes, and pixels are masked using the 18 February fSCA data. Refer to #45 for the new text better explaining the fSCA data's utility.

47. 3) What was SWE in the fSCA-masked areas? This is a measure of the uncertainty/error in the SWE extraction. It is a good parameter to calculate and report.

We calculated the ΔSWE in the fSCA-masked areas and reported the results in Section 3.1.

*Texted added reads: "As a first-order estimate of uncertainty within the technique, we calculated the $\Delta$SWE values for areas considered snow free by the 18 February fSCA data (Figure 4d). The $\Delta$SWE data from the three pairs (12--19, 19--26, and 12--26 February) were combined, and we report a snow free $\Delta$SWE mean value of -2.06 cm, an SD of 1.56 cm, and an IQR of 2.14 cm."*

48. L258 - change in SWE, not SWE

Updated to: *"InSAR phase differences produce a relative measurement of change in SWE..."*

49. L258-259 - Be more precise in your description, what does 'tether' mean in this context? Calibrating? Validating?

Tether updated to tie as per the recommendation in #36.

50. L282 - Your in situ measurement uncertainties on all the data need to be reported in the earlier sections where you describe the measurements. Reporting them in the results section for the first time happens a lot, so I'm not going to mention it any more, just check throughout.

See #26.

51. L263-264 - 'eight surrounding pixels' - did you exclude the snow pit pixel itself? I can understand why you might, but it should be made clear.

The snow pit pixel was included in the calculation for a total of nine pixels.

Text updated to: *"To account for error within GPS snow pit location $\Delta$SWE values for the snow pit pixel and the eight surrounding pixels were averaged. This averaged value was subtracted to obtain an absolute change. To calculate the cumulative $\Delta$SWE, the 12--19 February and 19--26 February were masked so only pixels that occurred in both scenes were considered and then added together."*

52. L263 - Geocoding of UAVSAR is not a problem - see Fore et al. You average to reduce the RANDOM errors, i.e., temporal decorrelation.

Fore, A. G., Chapman, B. D., Hawkins, B. P., Hensley, S., Jones, C. E., Michel, T. R., & Muellerschoen, R. J. (2015). UAVSAR Polarimetric Calibration. IEEE Transactions on Geoscience and Remote Sensing, 53(6), 3481-3491.

We removed the reference to error in geocoding of the InSAR data.

53. Section 3.1 - This entire section should be placed earlier in the paper, before or after discussing in situ data.

We moved Section 3.1 to the new section 2.3.1 'UAVSAR Data'.

54. L275 - what does 'preserved in phase unwrapping process' mean precisely?

'Preserved in the phase unwrapping process' means pixels that unwrapped successfully and contain phase information (not NA).

55. L276 - You need more description of the ISCE processing. Did you multilook? Filter? Coherence depends strongly on multilooking so that needs to be specified. Things like this are why I stated that the processing has to be done the same for all interferograms, otherwise comparisons don't mean much.

We added information on multilooking in Table 1 (see #25). As stated prior, all data processing has been standardized.

56. L281-282 - Re. 'the spatial variability in LWC...causes...' - this is assumed, not known via measurement. If you want to state this then justify it, maybe with a reference, and discuss other possible sources.

We removed the reference to LWC and updated the text: *"The spatial variability in backscatter values between acquisitions…"*

57. L283 - Re. 'in this riparian area' - Okay, this is why I want a map showing where the trees are and where they aren't. I can't check the images to verify what you are saying. Also, are you saying the LWC is varying only in the riparian area? What is the connection?

We added a canopy cover map in Figure 2g. We are not saying LWC only varies in this riparian area, and the text does not specify that.

Incorporating the recommendation from Review #2 (#22), the new text reads: *"This backscatter decrease is likely caused by snowpack LWC or subnivean surface water attenuating the radar signal."*

58. L283-284 - These 'artifacts' are definitely not the product of UAVSAR processing. They look to me like RFI (radio frequency interference) from external sources, e.g., FAA radars. I checked the UAVSAR products that I think you used and see that for the 2/12-19 pair those features show up in the coherence and the interferogram, but not in the unwrapped phase product. That is probably because additional spatial filtering is applied during UAVSAR standard phase unwrapping. My guess is that the details of the processing implemented in ISCE vs. used by the UAVSAR group are different, which is why the streaks don't show up in the 12-26 pair's coherence. Certainly, they can't all be due to the Feb. 19 acquisition since all the streaks in the 12-19 pair's coherence aren't in the 19-26 pair's coherence. Yet again I say to use exactly the same processing for all pairs.

Thank you for clarifying this.

We updated the text to: *"There are also horizontal streaks of low coherence and high backscatter within the images. These are likely a result of radio frequency interference (RFI) during the acquisitions."*

59. Table 3 -

**1)** Include the 12-26 pair in this table.

12-26 February pair added to Table 3.

**2)** The law of time reversal invariance tells us that the HV and VH backscatter from the surface must be the same. Differences reported in this table cannot relate to real differences in the surface scattering. Any differences in measured values come from errors in calibration, instrument cross talk, etc. Therefore, when making PolSAR products for UAVSAR, the average of HV and VH is used for a generic 'HV' product, the cross-polarization normalized radar cross section. You should do the same in your analysis. Otherwise, remove the values from the table since you only use the HH polarization in the end. The HV and VH products are provided separately for people working with very dark scenes who want to estimate the noise, and that is not the case here.

We removed HV and VH from the table.

New table and caption:

**Table 2.** UAVSAR unwrapped phase (UNW) and coherence statistics for the full scene (FS) and the study area (SA). UNW Loss (%) is the percentage of pixels lost in the unwrapping process.

| Pair | Polarization | FS Mean Coherence | SA Mean Coherence | FS UNW Loss (%) | VG UNW Loss (%) |
|------|------|------|------|------|------|
| 12–19 Feb. | HH | .53 | .50 | 9.4 | 7.7 |
| 12–19 Feb. | VV | .54 | .50 | 8.9 | 12.8 |
| 19–26 Feb. | HH | .55 | .52 | 5.0 | 4.1 |
| 19–26 Feb. | VV | .57 | .54 | 4.3 | 3.7 |
| 12–26 Feb. | HH | .50 | .48 | 8.3 | 6.6 |
| 12–26 Feb. | VV | .53 | .51 | 6.7 | 5.5 |

60. L288-289 - I don't agree with this statement. Most of the area around the VG meadow doesn't seem bad. It certainly phase unwrapped. If you are pointing out something important, it is definitely not obvious so add a figure and maybe quantify the difference.

This sentence was added to note more pixels are lost in the forested areas than in the open VG meadow.

Updated the text: *"The other source of low coherence and corresponding unwrapping pixel loss occurs on the forested hill slopes (outside of the blue dotted line) surrounding the VG meadow."*

61. L293 - values shown are not just in VG

See #62.

62. L298 - It would be valuable for you to quantify the difference between S and N facing slopes.

We added an analysis showing the difference between N and S facing slopes in Table 4. See

New text from section 3.1 InSAR ΔSWE:

[revised manuscript text omitted]

63. 3.2 Changes in SWE, general - You don't show the change in SWE for the BA area, but that is where the snowfall occurred. Please show that since you are using HQ as your reference. Being able to measure the SWE at BA is very important to your study, and if it is inconclusive or just very different then that too is important for estimating the uncertainty. Your study is the first of its kind and it is important to report both success and limitations so that future studies can improve on it. This is an opportunity to discuss in the conclusions what improvements need to be made (more snow, more frequent measurements, etc.)

Thank you for recommending this; it was a mistake on our end not to include the BA and HQ pits in the original submission. We added the BA and HQ pits SWE changes to Figure 11 and provided the updated figure in #75. The InSAR data agreed well with both pits, showing a smaller RMSE than the snow depth sensors.

We also appreciate the comments about our study. As we updated the text, we have focused on better quantification of the results and how these will impact future studies.

64. L303 - 'small storm' - all information about this and snowfall need to go in the data section, much earlier. This includes Fig. 14.

We moved Figure 14 to Figure 5c. Text about the small storm was added to the caption in Figure 5 as well. See #74.

65. L311 - Title needs to be Changes in SWE: InSAR vs. GPR and snow depth sensors, and hence you should combine sections 3.3 and the LAST SENTENCE of 3.4

Section 3.2 is now titled: *"InSAR vs. Snow Depth Sensors, Snow Pits, and GPR ΔSWE"*

66. L320 - I do not understand why this is 'likely' to be the MAXIMUM error. A 5% uncertainty in GPR can be positive or negative. Also, you didn't propagate all errors to get a maximum.

We removed this text and updated our GPR analysis in the Results and Discussion sections. See updated text in #82.

67. Section 3.4 -

**1)** All but the last sentence belongs much earlier in the data section

We moved the majority of the text to section 2.3.3.

**2)** What about discussing the results? The bias is much less than for GPR (Fig. 12). Some discussion of why is needed.

68. L338 - Why is the comparison only done for 12-26 pair? Shouldn't it be to the 19-26 pair? That is much closer in date to the optical data's dates and the snow fall event happened after 2/19. Compare to the 19-26 pair.

    We updated Figure 12 to compare the 19-26 February pair instead of the 12-26 pair. The new figure and caption are below.

[Figure]

*"**Figure 12.** (a) InSAR $\Delta$SWE between 19--26 February aggregated to the 30 m Landsat resolution, and (b) the change in Landsat fSCA between 18 February and 5 March. The color scale for (a) was changed to -5 to 5 cm to exemplify the patterns."*

69. L344 - The loss on S-facing slopes is not easy to see. Like I mentioned above, we need a map showing slope directions and, hopefully, some quantification of differences between S and N. Alternatively, you could show plots with just the values on the S slopes and just the values on the N slope.

We added a map of N/S facing slopes in Figure 2f and quantification of the SWE difference in Table 4. Section 3.1 was rewritten with the updated analysis and metrics. See #62 for the updated text and Table 4, and #72 for the updated histograms.

70. L353 - Like I mentioned above, we need a map showing where the forested areas are. You could overlay an outline on figures if you want to highlight something.

We added a canopy cover map in Figure 2g.

71. Fig. 7 - add points to show location of BA and HQ sites on the LH plots.

Added.

[Figure]

*"**Figure 3.** The unwrapped phase and coherence data for the (a) 12--19 February, (b) 19--26 February, and (c) 12--26 February InSAR pairs. (d) The amplitude data for the three UAVSAR flights. Both (a) and (c) were atmospherically corrected. The gray area in the phase data are pixels lost in the unwrapping processes. VG and Jemez River main channel are outlined by blue and red dotted lines, respectively. Triangles show the BA (red) and HQ (black) pits."*

72. Fig. 9 - Add a comparison of values inside VG vs. outside VG. Also make it clear exactly what area is covered by the histogram. I think it is the entire scene extent in Fig. 7 but it isn't clear.

We updated the figure to include histograms from the full study area, VG, north facing slopes, and south facing slopes. Section 3.1 has been updated to discuss these results. See #62.

Updated figure and caption:

[Figure]

*"**Figure 10.** The distribution of $\Delta$SWE values for the full study area, within VG, north facing, and south facing slopes. The top row displays the 12--19 and 12--26 February InSAR pairs (a--d), and the bottom row shows the 12--26 and 12--26 CM February InSAR pairs (d--h)."*

73. Fig. 10 - Instead of having V and H axes, have the 1:1 axis at 45deg. Also the colors for the two data sets are too similar.

We added a 1:1 line and changed the colors of the figure.

74. Fig. 11 - This goes much earlier, in the data section.

We moved this to Section 2.3.3 Snow Pit and Meteorlogic Data, combined it with Figure 14, and added insolation and wind speed. Per the recommendation #77, the plot time frame was extended, and vertical lines representing the acquisitions were added. New figure and caption below.

[Figure]

*"Figure 5. (a) A snow depth time series of the six CZO snow depth sensors (gray lines) (~3030 m) and HQ Met (black line) (2650 m). The gray-shaded area represents a small storm registered by the sensors on Redondo Peak. HQ Met and RP Met (red) (3231 m) time series of (b) average hourly temperature, (c) average hourly wind speed, (d) and average hourly incoming solar radiation (insolation) from 11 February to 6 March. The vertical blue dotted lines represent the three UAVSAR flights (12, 19, 26 February), and the vertical orange dotted lines represent the Landsat fSCA acquisitions (19 February & 5 March)."*

75. Fig. 12 - Your fit includes Feb 12-26 CM. That overcounts the short temporal baseline pairs relative to the independent Feb 12-26 pair. Remove the points for Feb 12-26 CM from the graph and recalculate the fit.

We removed Feb 12-26 CM and recalculated the fit for Figure 11. We also combined the GPR data and depth sensors/pit data into one figure. This uses the correct snow depth sensor geolocations.

[Figure]

*"**Figure 11.** (a) Comparing in situ SWE changes from the six CZO sensors (circles), HQ snow depth sensor (triangles), and the BA and HQ pits (stars) to InSAR-derived SWE changes for the three InSAR pairs. The depth sensor SWE error bars are derived from a*

*10 \% uncertainty from snow pit $\rho\_s$ measurements and $\pm$ 1 cm uncertainty from the ultrasonic depth sensors. (b) Comparing InSAR and GPR derived $\Delta$SWE from 12--26 February..”*

76. Fig. 13 - As said above, use 19-26 Feb instead. Maybe add a plot of delSWE vs. DelfSCA to show the correlation better.

    See #68. Given the differences in acquisition time and variable measure, we do not believe a plot scatter plot suits this situation.

77. Fig. 14 - Move to the data section. Extend plot out to the end of the fSCA data. Show fSCA (Landsat) acquisitions as vertical lines also. Add plots of wind speed and maybe direction since you mention that information in section 4.1.

    See #74.

78. L368-369 - Sentence 'While ...' belongs in the data section.

    This sentence to Section 2.3.3.

79. L373-374 - I think that your argument at change in LWC caused the InSAR decorrelation also supports the statement that water moved downslope.

    We agree with this.

80. L378 - Given the lack of snowfall, I think you could do a quantitative comparison, as I've suggested above.

    See #76.

81. L381 - 'remarkably well' - I would avoid qualitative statements like this. I think this is an overstatement and without more quantitative comparisons I would not draw this conclusion.

    We removed “remarkably well’ and added a quantitative comparison from Marshal et al. 2021.

82. L382-383 - I think that all your GPR data did was show a bias of -2 cm wrt InSAR, and that number disagreed with the snow depth measurements. Rather than general, and arguable, statements, use the in situ data to quantify the uncertainty and then use that to recommend different, i.e., more, InSAR measurements and different, ie., possibly more, in situ measurements in the future. Remember that this is one of the first studies and you can use it to justify and lay out a plan for the future. It wouldn't be a bad idea to reference

the targeted observables of the Decadal Survey and suggest missions for it, possibly in conjunction with NISAR.

We updated our GPR data analysis, as both reviewers did not agree with our original interpretation.

Updated text in Section 3.2 'InSAR vs. GPR, Snow Depth Sensors, and Snow Pits ΔSWE':

[revised manuscript text omitted]

83. L383 - I suggest adding the need for higher SWE during the measurements.

    See bold #82.

84. L392 - The calculation goes in earlier in the paper, but you should add a discussion of the conclusions re. LWC change here.

    We think this text is properly placed.

85. L401 - I don't agree that this is known to be the maximum bias. I don't think that was shown.

    We removed the reference to the maximum known bias from GPR for this study.

86. L405 - This is the first mention of 'lightly forested areas'. I don't think that this paper as written has really explored the difference between forested (heavy or light) vs. unforested in a quantitative way. In the event that you do, then discussion is justified

*We removed this text: "Furthermore, this method shows promise for lightly forested areas that were difficult to directly assess in our present study."*

87. L408 - could be water content also

*This is true. Dr. Meyer hypothesized it was a dry delay, but knowing absolutely is not possible.*

88. L416 - statement about Eppler study goes in data section, then can be referred to here.

*We added this statement to section Section 2.1 'InSAR for detecting SWE changes': "Recent studies from Eppler et al. (2022) and Leinss et al. (2015) found that error in density estimates only biases total SWE change by $< \sim 5\%$ for completely dry snow in a wide range of $\theta$ ($< 50^\circ$) and $\rho_s$ ($< 500$ km m $^{-3}$)."*

*And added discussion of it here: "Eppler et al. (2022) and Leinss et al. (2015) attributed $< \sim 5\%$ error to $\rho_s$ estimates. However, due to the known presence of LWC in the snowpack and the difference in timing between $\epsilon_s$ observations and UAVSAR flights, uncertainty is likely larger in our analysis."*

89. L431-432 - Tandem-X is X-band, so likely to measure canopy height. Why do you think that it will be better at getting the land surface than SRTM? I didn't read the entire reference but it would be worth stating quickly why if this statement is correct.

*Your statement about X-band in canopy is a very good point. We added: "..which does not show the same inaccuracies as SRTM for non-forested areas…"*

[Figure]

**Fig. 25.** Evolution of the Garzweiler mine in Germany. Optical images from © Google Earth from 2000 (a) and 2013 (b), respectively and corresponding SRTM (c) and TanDEM-X (d) DEMs.

*Figure from Rizzoli et al. (2017). A pattern of large mount features, similar to what we found in our scene, can be seen in the SRTM data (c) and the TanDEM-X (d).*

[Figure]

Copernicus 30 m screenshot from VG. Stream channels and terrain features are clear, with no large mounds. Our statements were based off checking the Copernicus DEM for many UAVSAR acquisition locations.

90. L454 - In fact, you showed that it WASN'T necessary to identify snow-covered pixels to do the atmospheric correction. Or at least that is my take-away from Fig. 5.

We can only be confident that there was an atmospheric delay because we compared snow free and snow covered pixels. If we were to have implemented our correction without delineating the snow free vs. snow covered pixels, there would be no way of knowing if snow cover changes were the main driver. While improbable within our study, this step builds the robustness of the correction methodology in various situations. Future situations will have large-scale snow cover and atmospheric phase signals in the scene. Using optical fSCA data allows us to understand the type of atmospheric delay better.

91. L454-455 - You never showed the difference between using a snow-on mask vs. not using a snow-on mask anywhere in this paper. I cannot therefore conclude that it is necessary. In fact, the great value in using that is in determining the uncertainty in your SWE measurement, which was not done. It should be.

See #45 for the updated fSCA justification. We show that 5% of pixels are snow free within the study area, and 30% within the whole UAVSAR for 18 February. This

multisensor approach is preparing for NISAR, which will have much larger acquisition areas.

See #47 where we report the fSCA dSWE statistics.

Below is an example of the dSWE data for 12-19 February.

[Figure]

This plot shows the snow masked data.

[Figure]

And this plot shows the areas considered snow free.

[Figure]

We added this text to the bottom of Section 3.1 InSAR ΔSWE:

*"As a first-order estimate of uncertainty within the technique, we calculated the $\Delta$SWE values for areas considered snow free by the 18 February fSCA data (Figure 4d). The $\Delta$SWE data from the three pairs (12--19, 19--26, and 12--26 February) were combined, and we report a snow free $\Delta$SWE mean value of -2.06 cm, an SD of 1.56 cm, and an IQR of 2.14 cm."*

We added this text to the bottom of Section 4.3:

*"Our results provide an initial evaluation of uncertainty of the InSAR-derived SWE changes by reporting the mean (-2.06 cm) and SD (1.56 cm) $\Delta$SWE values of snow free pixels. It is important to note that these pixels only represent about 5 \% of the study area, and much of the snow free area exists in densely forested regions where the fSCA uncertainty is greatest \citep{selkowitzUSGSLandsatSnow2017}. Section 4.3 outlines the continued work needed better to understand uncertainty within the InSAR SWE retrieval technique."*

92. L460 - Ditto above - need for this was not demonstrated to this reviewer. If you think that you showed it, then make it more obvious.

    See #45 and #47.

93. L468 - List delfSCA as a validation data set here.

    Text updated at LXXX to: "We then used in situ snow depth and density measurements, **$\Delta$fSCA**, and GPR SWE data to validate the InSAR SWE returns."

94. Data and code availability - general - Specific product names for everything you used needs to be provided. It is not enough to point to huge databases where someone searches and guesses what exactly you used.

    We added a supplement that provides specific product names for all spatial and in situ data used.

**Technical Corrections**

95. The entire manuscript needs to be read over to identify and correct errors in grammar, spelling, and language usage/wording. I only point out a few below, but there are lots of places that deserve more attention.

    We have edited the manuscript and corrected the grammar.

96. Everywhere - 'data' is plural, so data are, data were

Data updated to plural throughout the manuscript.

97. L36 - 1970s

Updated.

98. L71 - pairs from the Envisat…

Text updated at LXX to: *"...analyzed two InSAR pairs from the Envisat ASAR instrument…"*

99. L72 - lack of in situ

Updated.

100.   L76 - its

Updated.

101.   L104 - 'can calculate' needs to be 'can be used to calculate' or 'is related linearly to' L 140 - have two 'the'

'Can calculate' changed to 'can be used to calculate'.

102.   L 207 - correct, not corrected

The whole sentence has been updated.

103.   L335-336 - 'pixels they're located' -> deltaSWE

$\Delta$SWE added.

104.   L415 - in situ measured epsilon...

Updated.

We had hoped the SnowEx snow pit data set would be published by NSIDC – this is not the case. We added a citation for the SnowEx Hackweek database. This database isn't currently online, but hopefully, the data will be public on NSIDC soon.

**Supplementary Material**

**S1. List of Remote Sensing Data**

**UAVSAR Ground Projected Interferograms**
We used the ground projected (GRD) InSAR and PolSAR data from the 'alamos' flight line from three dates. This included HH and VV unwrapped phase (.unw), coherence (.cor), amplitude (.amp), and the DEM (.hgt). This data is available at the ASF DAAC (https://search.asf.alaska.edu/#/) or UAVSAR data portal (https://uavsar.jpl.nasa.gov/).

- 12–19 February
  alamos_35915_20005-003_20008-000_0007d_s01_L090_01
  https://uavsar.jpl.nasa.gov/cgi-bin/product.pl?jobName=alamos_35915_20005-003_20008-000_0007d_s01_L090_01#data

- 19–26 February
  alamos_35915_20008-000_20013-000_0007d_s01_L090_01
  https://uavsar.jpl.nasa.gov/cgi-bin/product.pl?jobName=alamos_35915_20008-000_20013-000_0007d_s01_L090_01#data

- 12–26 February
  UA_alamos_35915_20005-003_20013-000_0014d_s01_L090_01
  https://uavsar.jpl.nasa.gov/cgi-bin/product.pl?jobName=alamos_35915_20005-003_20013-000_0014d_s01_L090_01#data

**UAVSAR SLC Stack**
We also used an SLC stack to geolocate the new incidence angle data.

- 12, 19, and 26 February
  https://uavsar.jpl.nasa.gov/cgi-bin/product.pl?jobName=alamos_35915_04#data
  alamos_35915_20013_000_200226_L090HH_04_BU.ann
  alamos_35915_04_BU_s1_2x8.lkv
  alamos_35915_04_BU_s1_2x8.llh
  alamos_35915_04_BU.dop
  alamos_35915_20013_000_200226_L090HH_04_BU_s1_2x8.slc

**Landsat 8 fSCA**
These data are available through the USGS EarthExplorer data portal (https://earthexplorer.usgs.gov/). There are no specific product pages for these data.

- 5 March 2020
  Tile ID: LC08_CU_010012_20200305_20210504_02_SNOW

- 18 February 2020
  Tile ID: LC08_CU_010012_20200218_20210504_02_SNOW

**Landsat 8 C2 ARD Reflectance**
These data are available through the USGS EarthExplorer data portal
(https://earthexplorer.usgs.gov/). There are no specific product pages for these data.

- 5 March 2020
  Tile ID: LC08_CU_010012_20200305_20210504_02

- 18 February 2020
  Tile ID: LC08_CU_010012_20200218_20210504_02

**Lidar DEM**
- OpenTopography Jemez River Basin Snow-off LiDAR from July 2010
  https://doi.org/10.5069/G9RB72JV

**Canopy Cover**
- NLCD 2016 USFS Tree Canopy Cover (CONUS)
  https://www.mrlc.gov/data/nlcd-2016-usfs-tree-canopy-cover-conus

**S2. In Situ Data**
**Meteorologic Data**
We used data from two stations maintained by the Western Regional Climate Center (WRCC).

- Redondo (VC) New Mexico
  https://wrcc.dri.edu/cgi-bin/rawMAIN.pl?nvvrdd

- Valle Grande, New Mexico Weather Station
  https://wrcc.dri.edu/cgi-bin/rawMAIN.pl?nvvhvc

**GPR**
This data set contains two-way travel times from a ground penetrating radar survey conducted at
Jemez, New Mexico. Data were collected between 12 February 2020 and 04 March 2020 as part
of the SnowEx 2020 campaign.

- https://nsidc.org/data/snex20_j_unm_gpr/versions/1

---

## Editor Decision (ED1)

2023-03-23
Submission tc-2022-224

**Estimating snow accumulation and ablation with L-band InSAR**

Jack Tarricone et al.

Dear Dr. Tarricone, thank you for your submission to be considered for publication in The Cryosphere. The paper benefitted a very thorough review by the reviewers as well as a comment from the public. Although reviews highlighted major revisions, the quality of the research was underlined by the reviewers. The reviews suggested overall an improvement on the quantitative analysis on uncertainties paving the way toward future work. This was highlighted by both reviewers where specific effect of wetness on the SWE retrieval was questioned.

After my own reading of the paper, and the answer to reviewers, I fell the responses provided by the authors is well structured, justified and meet the reviewer's requirements. I am pleased to see a well-structured and detailed response that really consider all questions. This is a very thorough work by the authors. I therefore am confident that main concerns raised by the reviewers have been addressed and the paper can be published.

Regards,

Prof. Dr. Alexandre Langlois

Associate editor, *The Cryosphere*